# SHARP-Q: Spectral Hessian Alignment and Rectification for Post-training Quantization

**Menghao Lv** [1]  **Huiqiong Wang** [2]  **Li Sun** [2]  **Mingli Song** [1]

## Abstract

Post-training quantization (PTQ) suffers from severe accuracy degradation in ultra-low-bit regimes. To address this challenge, we propose SHARP-Q, a unified framework grounded in Information Geometry that aligns the quantization objective with the intrinsic Fisher geometry. Following a "Rectify-then-Approximate" strategy, SHARP-Q first preconditions the optimization landscape via Hessian-Aware Rectification (HAR) and subsequently approximates the rectified Fisher Information Matrix through Dynamic Fisher-Subspace Compensation (DFSC). Our empirical evaluations reveal a pivotal insight: precise geometric alignment enables hardware-friendly uniform quantizers to outperform specialized non-uniform designs. Extensive experiments across representative convolutional networks, Vision Transformers, and State Space Models confirm that SHARP-Q establishes new state-of-the-art results, achieving substantial accuracy gains in the challenging W2A2 and W3A3 settings.

## 1. Introduction

The deployment of over-parameterized deep neural networks on resource-constrained edge devices remains a significant challenge due to prohibitive memory footprints and inference latency. To mitigate these constraints, Post-Training Quantization (PTQ) has emerged as the prevailing paradigm for model compression. Unlike Quantization-Aware Training (QAT), which requires extensive retraining and access to full datasets, PTQ enables high-efficiency deployment using only a limited set of unlabeled calibration data. This makes PTQ particularly indispensable in scenarios involving data privacy concerns or where the massive computational cost of retraining is prohibitive.

The primary bottleneck in ultra-low-bit PTQ is the geometric misalignment between isotropic quantization objectives and the anisotropic optimization landscape. Conventional reconstruction methods typically rely on Mean Squared Error (MSE) (Wei et al., 2022) or diagonal Hessian approximations (Li et al., 2021); however, these approaches implicitly assume parameter independence, thereby failing to account for the complex cross-dimensional couplings essential for high-fidelity recovery. While some strategies incorporate curvature information, their effectiveness is often constrained by the underlying landscape's ill-conditioning. Attempting to optimize directly over such raw, ill-conditioned surfaces poses significant challenges for efficiently capturing critical sensitivities. Furthermore, to mitigate these geometric hurdles, some approaches necessitate structural compromises—such as altering native activation functions—thereby sacrificing the architectural universality fundamental to the PTQ paradigm. This methodological gap necessitates a unified framework capable of navigating complex curvature geometry without compromising the integrity of the original model.

To resolve these bottlenecks, we introduce SHARP-Q, a unified PTQ framework grounded in Information Geometry. Diverging from prior efforts that attempt to optimize over the raw, ill-conditioned error surface, SHARP-Q implements a systematic "Rectify-then-Approximate" strategy. Instead of passively accommodating the landscape's inherent anisotropy, our approach actively transforms the optimization geometry into a well-conditioned foundation through Hessian-Aware Rectification. This preconditioning ensures the accuracy of the subsequent Fisher Information Matrix approximation, allowing SHARP-Q to capture previously obscured critical sensitivities. By aligning the optimization geometry rather than altering the network structure, SHARP-Q enables uniform quantizers to achieve high-fidelity reconstruction, thereby preserving the structural integrity and deployment efficiency of the original model.

Our main contributions are summarized as follows:

**(i)** We establish a unified theoretical framework grounded in Information Geometry. By deriving the Fisher-KL Oracle,

[1] Zhejiang University, Hangzhou, China [2] Ningbo Global Innovation Center, Zhejiang University, Ningbo, China. Correspondence to: Huiqiong Wang <huiqiong_wang@zju.edu.cn>.

*Proceedings of the 43rd International Conference on Machine Learning*, Seoul, South Korea. PMLR 306, 2026. Copyright 2026 by the author(s).

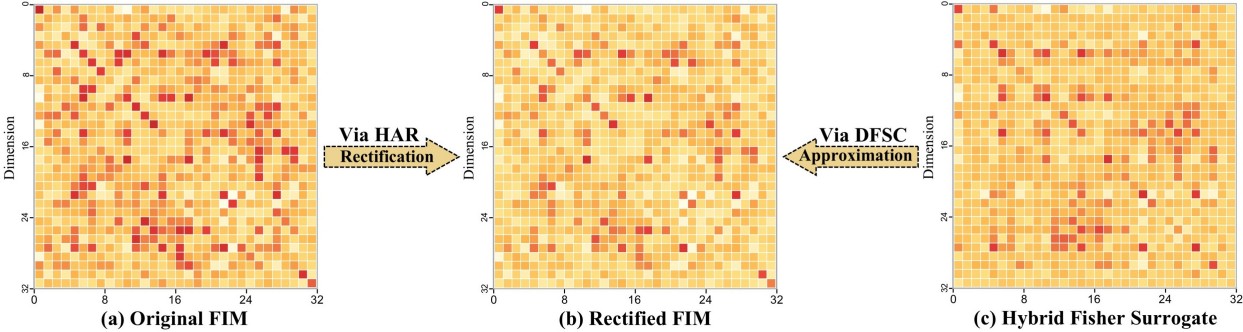

*Figure 1.* **Visualization of FIM Rectification and Approximation in SHARP-Q.** (a) *Original FIM:* The raw FIM exhibits severe anisotropy with dense off-diagonal correlations (highlighted regions). This complex geometry creates a fundamental mismatch for isotropic MSE and diagonal proxies, which inherently neglect these critical couplings. (b) *Rectified FIM:* HAR preconditions the landscape by dampening extreme cross-dimensional couplings, transforming the intractable geometry into a well-conditioned foundation for the subsequent stage. (c) *Hybrid Fisher Surrogate:* DFSC constructs a hybrid surrogate to capture critical sensitivities inaccessible to prior methods. The similarity between (b) and (c) demonstrates that the surrogate accurately approximates the rectified FIM, ensuring high-precision quantization.

we enable gradient-based estimation of the Fisher Information Matrix (FIM), which captures essential curvature information without the need for direct second-order derivative computation.

**(ii)** We operationalize the "Rectify-then-Approximate" strategy through two novel components: Hessian-Aware Rectification (HAR), which preconditions the ill-conditioned optimization landscape to create a well-conditioned foundation, and Dynamic Fisher-Subspace Compensation (DFSC), which subsequently captures critical sensitivity information typically lost in standard reconstruction objectives.

**(iii)** We conduct extensive evaluations across diverse convolutional networks, Vision Transformers, and State Space Models (Mamba), establishing new state-of-the-art performance. Our empirical results demonstrate that SHARP-Q enables hardware-friendly uniform quantizers to surpass even specialized non-uniform methods, particularly in the challenging ultra-low-bit regimes.

## 2. Related Work

Post-training quantization (PTQ) has established itself as a standard for data-efficient model compression. Early approaches like ACIQ (Banner et al., 2018) focused on analytical clipping, while AdaRound (Nagel et al., 2020) advanced this by formulating rounding as a quadratic optimization problem.

**Quantization for CNNs.** Block-wise reconstruction remains the dominant paradigm in CNN quantization. BRECQ (Li et al., 2021) pioneered the use of the Fisher Information Matrix (FIM) to approximate the Hessian, though it relies on block-diagonal assumptions. Subsequent frameworks such as QDrop (Wei et al., 2022) and PD-Quant (Liu et al., 2023) improved reconstruction through activation dropout and global prediction alignment, respectively.

More recently, CL-Calib (Shang et al., 2024) introduced contrastive learning to regularize calibration, albeit at the cost of increased computational complexity. These methods, however, remain fundamentally constrained by their reliance on isotropic MSE or diagonal proxies. By neglecting the complex cross-dimensional couplings inherent in the ill-conditioned optimization landscape, they encounter an inevitable bottleneck that prevents high-fidelity recovery in ultra-low-bit regimes.

**Quantization for ViTs.** The heavy-tailed distributions in Vision Transformers necessitate architecture-specific handling. Initial efforts like FQ-ViT (Lin et al., 2021) and PTQ4ViT (Yuan et al., 2022) designed specialized quantizers for LayerNorm and Softmax. Later research focused on smoothing outliers via reparameterization (RepQ-ViT (Li et al., 2023)) or suppressing oscillations (OASQ (Ma et al., 2024)). Most recently, efforts have shifted toward capturing curvature geometry. FIMA-Q (Wu et al., 2025a) explores Fisher approximations to guide quantization; however, it operates directly on the raw, ill-conditioned landscape, where dispersed spectral energy makes precise approximation computationally elusive. Similarly, APHQ-ViT (Wu et al., 2025b) proposes an Average Perturbation Hessian metric but necessitates structural compromises—replacing GELU with ReLU.

**Quantization for State Space Models.** Recent efforts have extended PTQ to state-space models like Mamba. For instance, MambaQuant (Xu et al., 2025) and Quamba (Chiang et al., 2025) address outlier amplification in selective scan mechanisms via variance-aligned rotations and Hadamard transforms, respectively. Similarly, SPR$^2$Q (Xin et al., 2026) designs specialized low-rank compensation for Mamba-based super-resolution. While highly effective, these approaches are inherently tailored to the unique structural properties of state-space models. In contrast, we demonstrate that SHARP-Q's geometric principles naturally generalize to these architectures, further confirming its universality.

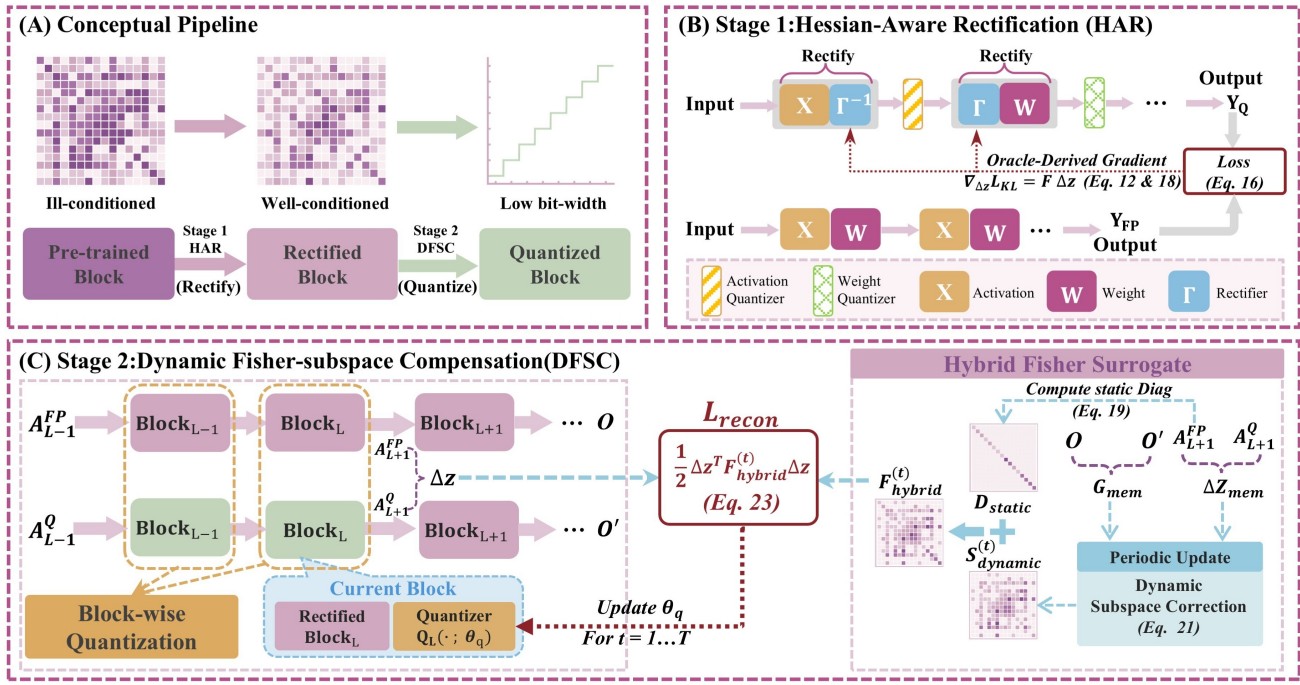

*Figure 2.* **Overview of the SHARP-Q Framework.** **(A)** *Conceptual Pipeline.* Illustrates our "Rectify-then-Approximate" strategy: transforming the ill-conditioned anisotropic landscape into a well-conditioned state to enable high-fidelity quantization. **(B)** *Stage 1: Hessian-Aware Rectification (HAR).* Learns a preconditioning operator to rectify the optimization landscape. This effectively mitigates complex cross-dimensional couplings, establishing a well-conditioned foundation for the subsequent stage. **(C)** *Stage 2: Dynamic Fisher-Subspace Compensation (DFSC).* Optimizes quantization parameters via a Hybrid Fisher Surrogate. By synergizing a robust static anchor with a dynamic subspace correction, DFSC efficiently captures critical sensitivities to guide precise quantization.

SHARP-Q departs from these paradigms by adopting a proactive "Rectify-then-Approximate" strategy. Prior efforts typically rely on isotropic objectives like MSE, resort to diagonal simplifications, or struggle with the inherent noise of ill-conditioned surfaces. In contrast, we first utilize HAR to transform the optimization landscape into a well-conditioned geometric basis. This preconditioning enables DFSC to capture critical sensitivities—including intricate off-diagonal interactions—with higher precision than previous proxies. By synergizing geometric rectification with precise Fisher approximation, SHARP-Q achieves state-of-the-art performance across diverse architectures.

## 3. Methodology

In this section, we introduce SHARP-Q, a framework designed to align the quantization process with the intrinsic Fisher geometry. As illustrated in Figure 2 and detailed in Algorithm 1, our "Rectify-then-Approximate" strategy follows a rigorous progression: we first model quantization as noise injection (§3.1) and derive the Fisher-KL Oracle (§3.2). Building on this theoretical basis, HAR (§3.3) geometrically preconditions the landscape to alleviate ill-conditioning, enabling DFSC (§3.4) to accurately approximate the rectified FIM via a Hybrid Surrogate for high-fidelity quantization.

### 3.1. Quantization as Noise Injection

We model post-training quantization as a structured noise injection process where perturbations interact with the optimization landscape's local curvature. For a pre-trained block with weights $\mathbf{W} \in \mathbb{R}^{C_{\text{out}} \times C_{\text{in}}}$ and activations $\mathbf{X} \in \mathbb{R}^{C_{\text{in}} \times N}$, the linear operation is:

$$\mathbf{z} = \mathbf{W}\mathbf{X}. \tag{1}$$

In Post-Training Quantization (PTQ), we map continuous values to discrete levels using a uniform quantizer $Q(\cdot; s, z_p)$. For an arbitrary tensor element $v$, the quantized version $\hat{v}$ is:

$$\hat{v} = Q(v; s, z_p) \triangleq s \cdot \text{clamp}\left(\left\lfloor \frac{v}{s} \right\rceil + z_p; 0, 2^b - 1\right) - s \cdot z_p, \tag{2}$$

where $b$ is the bit-width, $s \in \mathbb{R}_+$ is the scaling factor, $z_p \in \mathbb{Z}$ is the zero-point, $\lfloor \cdot \rceil$ denotes rounding-to-nearest, and clamp restricts values to the integer range. This introduces an irreversible error. Let $\boldsymbol{\epsilon}_w$ and $\boldsymbol{\epsilon}_x$ denote noise vectors for weights and inputs. The quantized output $\hat{\mathbf{z}}$ deviates from $\mathbf{z}$:

$$\hat{\mathbf{z}} = (\mathbf{W} + \boldsymbol{\epsilon}_w)(\mathbf{X} + \boldsymbol{\epsilon}_x). \tag{3}$$

To facilitate curvature analysis, we model the output perturbation $\Delta\mathbf{z}$ via its linearized approximation:

$$\Delta\mathbf{z} \triangleq \hat{\mathbf{z}} - \mathbf{z} \approx \mathbf{W}\boldsymbol{\epsilon}_x + \boldsymbol{\epsilon}_w\mathbf{X}. \tag{4}$$

**Algorithm 1** SHARP-Q for Block-wise Quantization

1: **Input:** FP model $\mathcal{M}_{\text{fp}}$, Calibration data $\mathcal{D}_{\text{calib}}$, Subspace rank $r$, HAR iterations $T_{\text{har}}$, Quantization reconstruction iterations $T_{\text{quant}}$.
2: **Output:** Optimized quantized model $\mathcal{M}_{\text{q}}$.
3: Initialize quantized model: $\mathcal{M}_{\text{q}} \leftarrow \mathcal{M}_{\text{fp}}$.
4: Calculate subspace update interval $\tau = \lfloor T_{\text{quant}}/r \rfloor$.
5: **for** each block $\mathcal{B}$ in $\mathcal{M}_{\text{q}}$ **do**
6:   *# Stage 1: HAR*
7:   Initialize preconditioner parameters $\mathbf{P}$.
8:   **for** $t = 1$ **to** $T_{\text{har}}$ **do**
9:     Compute HAR loss $\mathcal{L}_{\text{HAR}}$ using $\mathcal{D}_{\text{calib}}$ (Eq. 16).
10:    Update $\mathbf{P}$ via the Fisher-KL Oracle $\nabla \mathcal{L}_{\text{KL}} \approx \mathbf{F}\Delta\mathbf{z}$ (Eqs. 12, 18).
11:  **end for**
12:  Fix $\mathbf{\Gamma}^* = \exp(\mathbf{P})$ and fuse into weights: $\tilde{\mathcal{B}} \leftarrow$ Rectify$(\mathcal{B}, \mathbf{\Gamma}^*)$.
13:  *# Stage 2: DFSC*
14:  Compute static diagonal anchor $\mathbf{D}_{\text{static}}$ on $\tilde{\mathcal{B}}$ (Eq. 19).
15:  Initialize quantization parameters $\Theta_q$ (AdaRound weights $\mathbf{V}$, step sizes $s$).
16:  **for** $t = 1$ **to** $T_{\text{quant}}$ **do**
17:    **if** $t \bmod \tau = 0$ **or** $t = 1$ **then**
18:      Update dynamic correction matrix $\mathbf{S}_{\text{dynamic}}^{(t)}$ (Eqs. 20, 21).
19:      Construct hybrid surrogate $\mathbf{F}_{\text{hybrid}}^{(t)}$ (Eq. 22).
20:    **end if**
21:    Update $\Theta_q$ by minimizing $\mathcal{L}_{\text{recon}}$ (Eq. 23).
22:  **end for**
23:  Finalize quantized block $\tilde{\mathcal{B}}$ with $\Theta_q$ and update model $\mathcal{M}_{\text{q}}$.
24: **end for**
25: **return** $\mathcal{M}_{\text{q}}$

---

**Remark:** We omit higher-order terms (e.g., $\epsilon_w \epsilon_x$), which vanish under the standard small-perturbation assumption in PTQ analysis.

### 3.2. Geometric Misalignment and the Fisher-KL Oracle

We aim to minimize the expected performance degradation induced by $\Delta\mathbf{z}$:

$$\min \mathbb{E}_{\mathbf{x} \sim \mathcal{D}_{\text{data}}} [\Delta\mathcal{L}] = \mathbb{E}_{\mathbf{x} \sim \mathcal{D}_{\text{data}}} [\mathcal{L}(\mathbf{z} + \Delta\mathbf{z}) - \mathcal{L}(\mathbf{z})]. \quad (5)$$

**From Loss Landscape to Fisher Geometry.** Assuming convergence ($\nabla_{\mathbf{z}}\mathcal{L} \rightarrow \mathbf{0}$), the degradation is dominated by the second-order term:

$$\Delta\mathcal{L} = \underbrace{\nabla_{\mathbf{z}}\mathcal{L}(\mathbf{z})^\top \Delta\mathbf{z}}_{\approx 0} + \frac{1}{2}\Delta\mathbf{z}^\top \mathbf{H}\Delta\mathbf{z} + \mathcal{O}(\|\Delta\mathbf{z}\|^3)$$
$$\approx \frac{1}{2}\Delta\mathbf{z}^\top \mathbf{H}\Delta\mathbf{z}, \quad (6)$$

where $\mathbf{H} = \nabla_{\mathbf{z}}^2 \mathcal{L}(\mathbf{z})$. Explicitly computing $\mathbf{H}$ ($N \times N$) is computationally prohibitive.

To resolve this intractability, we utilize the property that for standard task losses formulated as Negative Log-Likelihood (NLL), the expected Hessian coincides with the Fisher Information Matrix (FIM) $\mathbf{F}$ at convergence (i.e., where the expected gradient vanishes) (Martens, 2020):

$$\mathbb{E}_{\mathbf{y}}[\mathbf{H}] = \mathbf{F} \triangleq \mathbb{E}_{\mathbf{y} \sim p(\cdot|\mathbf{z})} \left[ \nabla_{\mathbf{z}} \log p(\mathbf{y}|\mathbf{z}) \nabla_{\mathbf{z}} \log p(\mathbf{y}|\mathbf{z})^\top \right]. \quad (7)$$

Substituting this into Eq. (6), the target becomes minimizing the Fisher-induced Mahalanobis distance:

$$\min \mathcal{D}_{\mathbf{F}}(\mathbf{z}, \hat{\mathbf{z}}) = \frac{1}{2}\Delta\mathbf{z}^\top \mathbf{F}\Delta\mathbf{z}. \quad (8)$$

**Geometric Misalignment of Isotropic Objectives.** Standard MSE minimization ($\mathcal{L}_{\text{MSE}} = \|\Delta\mathbf{z}\|^2$) implicitly assumes isotropic curvature ($\mathbf{F} \propto \mathbf{I}$). Since deep networks exhibit highly anisotropic $\mathbf{F}$, a fundamental mismatch exists:

$$\Delta\mathcal{L} \approx \frac{1}{2}\Delta\mathbf{z}^\top \mathbf{F}\Delta\mathbf{z} \neq \frac{1}{2}\Delta\mathbf{z}^\top \mathbf{I}\Delta\mathbf{z} \propto \mathcal{L}_{\text{MSE}}. \quad (9)$$

This mismatch implies that isotropic objectives such as MSE fail to account for critical sensitivity information, often leading to suboptimal quantization on anisotropic error surfaces.

To access $\mathbf{F}$ geometry without explicit storage, we leverage the asymptotic property of the KL divergence between the output distributions of the FP32 teacher ($\mathcal{T}$) and the quantized student ($\mathcal{S}$):

$$D_{\text{KL}}(\mathcal{T} \| \mathcal{S}) = \sum_{\mathbf{y} \in \mathcal{Y}} p(\mathbf{y}|\mathbf{z}) \log \frac{p(\mathbf{y}|\mathbf{z})}{p(\mathbf{y}|\mathbf{z} + \Delta\mathbf{z})}. \quad (10)$$

A second-order expansion reveals the connection to the Fisher metric:

$$\begin{aligned} \mathcal{L}_{\text{KL}} &\triangleq D_{\text{KL}}(\mathcal{T} \| \mathcal{S}) \\ &= \underbrace{D_{\text{KL}}(\mathbf{0})}_{0} + \underbrace{\nabla D_{\text{KL}}(\mathbf{0})^\top}_{\mathbf{0}} \Delta\mathbf{z} \\ &\quad + \frac{1}{2}\Delta\mathbf{z}^\top \mathbf{F}\Delta\mathbf{z} + \mathcal{O}(\|\Delta\mathbf{z}\|^3). \end{aligned} \quad (11)$$

This expansion confirms that minimizing $\mathcal{L}_{\text{KL}}$ is equivalent to minimizing the Fisher-weighted quadratic form. Crucially, the gradient of $\mathcal{L}_{\text{KL}}$ serves as an exact Fisher-KL Oracle:

$$\nabla_{\hat{\mathbf{z}}} \mathcal{L}_{\text{KL}} \approx \frac{\partial}{\partial \Delta\mathbf{z}} \left( \frac{1}{2}\Delta\mathbf{z}^\top \mathbf{F}\Delta\mathbf{z} \right) = \mathbf{F}\Delta\mathbf{z}. \quad (12)$$

While this Fisher-KL Oracle enables efficient access to Fisher information, the intrinsic geometric ill-conditioning of $\mathbf{F}$ persists. Building upon this insight, SHARP-Q first employs HAR to precondition the landscape, establishing a well-conditioned basis for subsequent DFSC approximation.

### 3.3. Stage 1: Hessian-Aware Rectification

The ill-conditioned geometry of the optimization landscape amplifies the model's sensitivity to quantization noise. HAR addresses this by establishing a well-conditioned basis through the transformation of the linear layer with a learnable Preconditioning Operator $\mathbf{\Gamma} \in \mathbb{R}^{C_{\text{in}} \times C_{\text{in}}}$:

$$\mathbf{z} = (\mathbf{W}\mathbf{\Gamma})(\mathbf{\Gamma}^{-1}\mathbf{X}). \tag{13}$$

Importantly, this rectification is deployment-friendly: the forward term fuses into the weights ($\tilde{\mathbf{W}} = \mathbf{W}\mathbf{\Gamma}$), while the inverse term $\mathbf{\Gamma}^{-1}$ is seamlessly handled via efficient operator fusion or lightweight pre-scaling (detailed in Appendix D).

Direct optimization of $\mathbf{\Gamma}$ is numerically unstable due to positive definiteness constraints. We thus employ a Logarithmic Parameterization $\mathbf{P}$ to enforce stability:

$$\mathbf{\Gamma}(\mathbf{P}) = \exp(\mathbf{P}) \triangleq \text{diag}(e^{p_1}, \dots, e^{p_{C_{\text{in}}}}), \quad \mathbf{P} \in \mathbb{R}^{C_{\text{in}}}. \tag{14}$$

To further guarantee the validity of local Taylor approximations during the update trajectory, we impose a strict trust region in the parameter space:

$$d_g^2(\mathbf{\Gamma}, \mathbf{\Gamma}_{\text{init}}) = \| \log \mathbf{\Gamma} - \log \mathbf{\Gamma}_{\text{init}} \|_F^2 = \| \mathbf{P} - \mathbf{P}_{\text{init}} \|_2^2. \tag{15}$$

Synthesizing the geometric curvature alignment with these regularization constraints, we formulate the rectification objective for this stage as:

$$\mathcal{L}_{\text{HAR}}(\mathbf{P}) = \mathbb{E}\left[ \underbrace{\mathcal{L}_{\text{KL}}(\mathcal{T} \| \mathcal{S}(\mathbf{P}))}_{\text{Spectral Conditioning}} + \frac{\lambda}{2} \underbrace{\| \mathbf{P} - \mathbf{P}_{\text{init}} \|_2^2}_{\text{Trust Region}} \right]. \tag{16}$$

Geometrically, this constraint anchors the optimization within the local validity of the Fisher-KL Oracle. Since Eq. 12 relies on a second-order Taylor expansion, the penalty prevents $\mathbf{P}$ from drifting into unstable regimes where the Fisher matrix ceases to be a reliable proxy, thereby ensuring optimization stability.

Applying the chain rule to $\mathcal{L}_{\text{HAR}}$ with respect to $\mathbf{P}$ yields the gradient:

$$\frac{\partial \mathcal{L}_{\text{HAR}}}{\partial \mathbf{P}} = \left( \frac{\partial \mathcal{L}_{\text{KL}}}{\partial \hat{\mathbf{z}}} \frac{\partial \hat{\mathbf{z}}}{\partial \mathbf{\Gamma}} \frac{\partial \mathbf{\Gamma}}{\partial \mathbf{P}} \right)^{\top} + \lambda(\mathbf{P} - \mathbf{P}_{\text{init}}). \tag{17}$$

Crucially, the gradient term $\nabla_{\hat{\mathbf{z}}} \mathcal{L}_{\text{KL}}$ in Eq. (17) naturally functions as our **Fisher-KL Oracle**, providing a stochastic estimate of the Fisher-vector product: $\nabla_{\hat{\mathbf{z}}} \mathcal{L}_{\text{KL}} \approx \mathbf{F}\Delta\mathbf{z}$ (as established in Eq. 12).

$$\mathbf{P}_{t+1} \leftarrow \mathbf{P}_t - \eta \left[ \left( \frac{\partial \mathbf{\Gamma}}{\partial \mathbf{P}} \right)^{\top} \left( \frac{\partial \hat{\mathbf{z}}}{\partial \mathbf{\Gamma}} \right)^{\top} (\mathbf{F}\Delta\mathbf{z}) + \lambda(\mathbf{P}_t - \mathbf{P}_{\text{init}}) \right]. \tag{18}$$

**Proposition 3.1** (Monotonic Spectral Compression). *Subject to the trust-region constraint, the KL-driven update (Eq. 18) monotonically minimizes the trace of the rectified curvature energy, $\mathbb{E}[\Delta\mathbf{z}^{\top} \mathbf{F}\Delta\mathbf{z}]$, effectively compressing the spectral envelope of the optimization landscape (see Appendix B for the derivation).*

Unlike methods relying on statistical heuristics to suppress outliers (e.g., RepQ-ViT (Li et al., 2023), OASQ (Ma et al., 2024)), HAR optimizes the geometry guided by the Fisher Information Matrix. While prior approaches implicitly equate smoothing activation magnitudes with error reduction, our strategy directly targets the underlying optimization curvature. As derived in Appendix B, minimizing the weighted trace implicitly dampens cross-dimensional couplings. This ensures that the rectification is governed by intrinsic geometric sensitivity rather than distributional statistics, establishing a well-conditioned foundation, thereby paving the way for the high-fidelity approximation in the subsequent stage.

### 3.4. Stage 2: Dynamic Fisher-Subspace Compensation

Building upon the rectified geometry established by HAR, we propose DFSC to construct a **Hybrid Fisher Surrogate** that accurately approximates the rectified FIM. This surrogate synergizes a robust static diagonal anchor with a dynamic subspace correction to capture critical sensitivities. Serving as the objective for quantization reconstruction, this proxy minimizes noise impacts where the model is most sensitive.

**Static Diagonal Anchor.** We first construct a robust diagonal baseline using the standard empirical Fisher diagonal to capture element-wise sensitivity:

$$\mathbf{D}_{\text{static}} \triangleq \text{diag}\left( \mathbb{E}_{\mathbf{y} \sim p(\cdot|\mathbf{z})}\left[ \nabla_{\mathbf{z}} \log p(\mathbf{y}|\mathbf{z}) \odot \nabla_{\mathbf{z}} \log p(\mathbf{y}|\mathbf{z}) \right] \right). \tag{19}$$

This term serves as a global stabilizer. While it cannot capture cross-dimensional interactions, it provides a conservative, full-rank estimate of the loss curvature, preventing the surrogate from overfitting to the low-dimensional subspace and ensuring numerical stability during optimization.

**Dynamic Subspace Correction.** To circumvent the prohibitive $\mathcal{O}(N^2)$ computational overhead associated with explicitly constructing the full $N \times N$ Fisher matrix, we employ a data-driven strategy to construct the subspace correction using historical perturbation tracking. During each periodic update phase (every $\tau$ iterations), we extract the batch-averaged absolute magnitude of the instantaneous output perturbation, yielding a compact sensitivity vector $\delta_i \in \mathbb{R}^N$. By appending a new vector $\delta_i$ exclusively at these periodic intervals, we progressively construct the historical perturbation matrix $\Delta\mathbf{Z}_{\text{mem}} = [\delta_1, \dots, \delta_r] \in \mathbb{R}^{N \times r}$. We then compute their corresponding Fisher responses $\mathbf{G}_{\text{mem}} \in \mathbb{R}^{N \times r}$

*Table 1.* Comparison with state-of-the-art PTQ methods on ImageNet (ViTs). We report Top-1 accuracy (%). The best results are **bolded**. **Optim.** denotes whether the method involves optimization-based reconstruction. **S-Quant** and **G-Quant** represent the specialized quantizers used for post-Softmax and post-GELU activations, respectively. '*' indicates results reproduced using the official code. Abbreviations for specialized quantizers: **TUQ** (Twin-Uniform), **Log2** (Logarithm-based), **MPQ** (Matthew-effect Preserving), **GUQ** (Groupwise Uniform), **SULQ** (Shift-Uniform-Log2), and **TanQ** (Tangent Quantizer).

| Method | Optim. | S-Quant | G-Quant | Bits (W/A) | ViT-S | ViT-B | DeiT-T | DeiT-S | Swin-S |
|---|---|---|---|---|---|---|---|---|---|
| Full Prec. | - | - | - | 32/32 | 81.39 | 84.54 | 72.21 | 79.85 | 83.23 |
| PTQ4ViT (Yuan et al., 2022) | × | TUQ | TUQ | 4/4 | 42.57 | 30.69 | 36.96 | 34.08 | 76.09 |
| APQ-ViT (Ding et al., 2022) | × | MPQ | Uniform | 4/4 | 47.95 | 41.41 | 47.94 | 43.55 | 77.15 |
| RepQ-ViT (Li et al., 2023) | × | Log2 | Uniform | 4/4 | 65.05 | 68.48 | 57.43 | 69.03 | 79.45 |
| ERQ (Zhong et al., 2024) | × | Log2 | Uniform | 4/4 | 68.91 | 76.63 | 60.29 | 72.56 | 80.74 |
| IGQ-ViT (Moon et al., 2024) | × | GUQ | GUQ | 4/4 | 73.61 | 79.32 | 62.45 | 74.66 | 80.98 |
| I&S-ViT (Zhong et al., 2023) | ✓ | SULQ | Uniform | 4/4 | 74.87 | 80.07 | 65.21 | 75.81 | 81.17 |
| DopQ-ViT (Yang et al., 2024) | ✓ | TanQ | Uniform | 4/4 | 75.69 | 80.95 | 65.54 | 75.84 | 81.71 |
| OASQ (Ma et al., 2024) | ✓ | Uniform | Uniform | 4/4 | 72.88 | 76.59 | 66.31 | 76.00 | 81.02 |
| FIMA-Q* (Wu et al., 2025a) | ✓ | Uniform | Uniform | 4/4 | 76.32 | 82.92 | 66.91 | 76.75 | 81.67 |
| APHQ-ViT (Wu et al., 2025b) | ✓ | Uniform | Uniform | 4/4 | 76.07 | 82.41 | 66.66 | 76.40 | 81.81 |
| AdaLog (Wu et al., 2024) | ✓ | AdaLog | AdaLog | 4/4 | 76.83 | 82.79 | 66.72 | 76.16 | 81.67 |
| **SHARP-Q (Ours)** | ✓ | **Uniform** | **Uniform** | 4/4 | **77.34** | **83.25** | **67.13** | **76.86** | **81.96** |
| PTQ4ViT (Yuan et al., 2022) | × | TUQ | TUQ | 3/3 | 0.10 | 0.10 | 3.50 | 0.10 | 28.69 |
| RepQ-ViT (Li et al., 2023) | × | Log2 | Uniform | 3/3 | 0.10 | 0.10 | 0.10 | 0.10 | 0.10 |
| I&S-ViT (Zhong et al., 2023) | ✓ | SULQ | Uniform | 3/3 | 45.16 | 63.77 | 41.52 | 55.78 | 74.20 |
| DopQ-ViT (Yang et al., 2024) | ✓ | TanQ | Uniform | 3/3 | 54.72 | 65.76 | 44.71 | 59.26 | 74.77 |
| AdaLog (Wu et al., 2024) | ✓ | AdaLog | AdaLog | 3/3 | 62.50 | 76.73 | 48.78 | 63.24 | 76.01 |
| FIMA-Q* (Wu et al., 2025a) | ✓ | Uniform | Uniform | 3/3 | 62.67 | 77.11 | 55.26 | 69.17 | 76.72 |
| APHQ-ViT (Wu et al., 2025b) | ✓ | Uniform | Uniform | 3/3 | 63.17 | 76.31 | 55.42 | 68.76 | 76.10 |
| **SHARP-Q (Ours)** | ✓ | **Uniform** | **Uniform** | 3/3 | **66.93** | **77.86** | **56.14** | **69.73** | **77.16** |

column-wise via the Fisher-KL Oracle:

$$\mathbf{G}_{\text{mem}} \triangleq \nabla_{\Delta \mathbf{z}_{\text{mem}}} \mathcal{L}_{\text{KL}} \approx \mathbf{F} \Delta \mathbf{Z}_{\text{mem}}. \qquad (20)$$

Directly aggregating these sequential vectors would introduce spectral redundancy because the tracked perturbations $\delta_i$ are non-orthogonal. We resolve this geometric overlap by incorporating the subspace correlation matrix $\mathbf{C}_{\text{sub}} \triangleq (\Delta \mathbf{Z}_{\text{mem}})^\top \Delta \mathbf{Z}_{\text{mem}} \in \mathbb{R}^{r \times r}$. This matrix captures the pairwise inner products of the historical directions. Inverting it effectively decorrelates the subspace basis, ensuring that each historical direction contributes independently. The **Dynamic Subspace Correction** is formally defined as:

$$\mathbf{S}_{\text{dynamic}}^{(t)} \triangleq \mathbf{G}_{\text{mem}}(\mathbf{C}_{\text{sub}})^{-1}(\mathbf{G}_{\text{mem}})^\top. \qquad (21)$$

**Hybrid Fisher Surrogate.** The Hybrid Surrogate is constructed by synergizing the static anchor with this dynamic correction, scaled by a balancing coefficient $\gamma$:

$$\mathbf{F}_{\text{hybrid}}^{(t)} = \mathbf{D}_{\text{static}} + \gamma \cdot \mathbf{S}_{\text{dynamic}}^{(t)}. \qquad (22)$$

Owing to the well-conditioned geometry established in Stage 1, a minimal subspace dimension $r$ (e.g., $r = 5$) is sufficient to capture the dominant curvature, enabling efficient $\mathcal{O}(Nr)$ computation.

**Proposition 3.2** (Spectral Alignment). *For perturbations within the dominant error subspace, the dynamic correction $\mathbf{S}_{\text{dynamic}}^{(t)}$ shares the principal eigenspace of the true Fisher Information Matrix (see Appendix C). This spectral alignment ensures that SHARP-Q prioritizes suppressing quantization noise along high-curvature directions, while the diagonal anchor maintains global full-rank stability.*

**Quantization Objective.** Finally, we optimize quantization parameters $\Theta_q$ by minimizing the surrogate reconstruction loss:

$$\min_{\Theta_q} \mathcal{L}_{\text{recon}} = \tfrac{1}{2} \Delta \mathbf{z}^\top \mathbf{F}_{\text{hybrid}}^{(t)} \Delta \mathbf{z}. \qquad (23)$$

## 4. Experiments

### 4.1. Experimental Setup

**Datasets and Models.** We evaluate SHARP-Q across a diverse range of architectures on standard vision benchmarks. For image classification on ImageNet (Deng et al., 2009), our evaluated **Vision Transformers** include ViT (Dosovitskiy, 2020), DeiT (Touvron et al., 2021), and Swin Transformer (Liu et al., 2021), utilizing pre-trained weights from the *timm* library. For **CNN architectures**, we select ResNet (He et al., 2016), MobileNetV2 (Sandler et al., 2018), RegNet (Radosavovic et al., 2020), and MNasNet (Tan et al., 2019), with checkpoints sourced from the BRECQ reposi-

*Table 2.* Comparison with state-of-the-art PTQ methods on ImageNet (CNNs). We report Top-1 accuracy (%). The best results are **bolded**. We abbreviate model names: **MbV2** (MobileNetV2), **RegX-600** (RegNetX-600MF), and **RegX-3.2** (RegNetX-3.2GF).

| Method | Bits (W/A) | ResNet-18 | ResNet-50 | MbV2 | RegX-600 | RegX-3.2 | MNasNet |
|---|---|---|---|---|---|---|---|
| Full Prec. | 32/32 | 71.01 | 76.63 | 72.62 | 73.52 | 78.46 | 76.52 |
| ACIQ-Mix (Banner et al., 2018) | 4/4 | 67.00 | 73.80 | - | - | - | - |
| LAPQ (Nahshan et al., 2021) | 4/4 | 60.30 | 70.00 | 49.70 | 57.71 | 55.89 | 65.32 |
| Bit-Split (Wang et al., 2020) | 4/4 | 67.00 | 73.80 | - | - | - | - |
| AdaRound (Nagel et al., 2020) | 4/4 | 67.96 | 73.88 | 61.52 | 68.20 | 73.85 | 68.86 |
| QDrop (Wei et al., 2022) | 4/4 | 69.17 | 75.15 | 68.07 | 70.91 | 76.40 | 72.81 |
| PD-Quant (Liu et al., 2023) | 4/4 | 69.30 | 75.09 | 68.33 | 71.04 | 76.57 | 73.30 |
| CL-Calib (Shang et al., 2024) | 4/4 | 69.41 | **75.38** | 68.56 | 71.38 | 76.40 | 73.60 |
| **SHARP-Q (Ours)** | 4/4 | **69.61** | 75.25 | **68.79** | **71.57** | **77.04** | **73.78** |
| LAPQ (Nahshan et al., 2021) | 2/4 | 0.18 | 0.14 | 0.13 | 0.17 | 0.12 | 0.18 |
| AdaRound (Nagel et al., 2020) | 2/4 | 0.11 | 0.12 | 0.15 | - | - | - |
| QDrop (Wei et al., 2022) | 2/4 | 64.57 | 70.09 | 53.37 | 63.18 | 71.96 | 63.23 |
| PD-Quant (Liu et al., 2023) | 2/4 | 65.07 | 70.92 | 55.27 | 64.00 | 72.43 | 63.33 |
| CL-Calib (Shang et al., 2024) | 2/4 | 65.14 | 70.92 | 55.63 | 64.50 | 72.82 | **63.46** |
| **SHARP-Q (Ours)** | 2/4 | **65.16** | **70.94** | **55.84** | **64.69** | **73.10** | 63.44 |
| QDrop (Wei et al., 2022) | 4/2 | 57.56 | 63.26 | 17.30 | 49.73 | 62.00 | 34.12 |
| PD-Quant (Liu et al., 2023) | 4/2 | 58.65 | 64.18 | 20.40 | 51.29 | 62.76 | 38.89 |
| CL-Calib (Shang et al., 2024) | 4/2 | 59.03 | 65.12 | 22.77 | 52.35 | 63.53 | 40.80 |
| **SHARP-Q (Ours)** | 4/2 | **60.40** | **65.48** | **24.78** | **56.93** | **67.88** | **41.83** |
| BRECQ (Li et al., 2021) | 2/2 | 42.54 | 29.01 | 0.24 | 3.58 | 3.62 | 0.61 |
| AdaQuant (Hubara et al., 2021) | 2/2 | 0.11 | 0.12 | 0.15 | - | - | - |
| QDrop (Wei et al., 2022) | 2/2 | 51.42 | 55.45 | 10.28 | 39.01 | 54.38 | 23.59 |
| PD-Quant (Liu et al., 2023) | 2/2 | 53.08 | 56.98 | 14.17 | 40.92 | 55.13 | 28.03 |
| CL-Calib (Shang et al., 2024) | 2/2 | 54.45 | 58.30 | 17.70 | 42.19 | 56.39 | 30.34 |
| **SHARP-Q (Ours)** | 2/2 | **56.74** | **60.54** | **20.81** | **46.89** | **61.23** | **32.26** |

tory (Li et al., 2021). Furthermore, to validate architectural universality, we extend our evaluation to **State Space Models**, specifically focusing on MambaIRv2-light (Guo et al., 2025) evaluated on standard Image Super-Resolution benchmarks (e.g., Set5, Urban100).

**Implementation Details.** All experiments are conducted using PyTorch on 4 NVIDIA A6000 GPUs. For CNNs and ViTs, we utilize 1,024 randomly sampled images from the ImageNet training set for calibration. We adopt channel-wise uniform quantizers for weights and layer-wise uniform quantizers for activations. Following standard practices (Wei et al., 2022; Wu et al., 2025b), we perform block-wise reconstruction for 20,000 iterations (batch size 32), with learning rates of $3e-3$ for weights and $4e-5$ for activation step sizes.

For the SHARP-Q specific components, the HAR stage is executed for 200 iterations (learning rate $3e-5$, batch size 64). In the subsequent DFSC stage, we employ a Fisher subspace rank of $r = 5$ and set the subspace update interval to $\tau = 20,000/r$. Following standard practices in ViT quantization (Wu et al., 2025b), we evaluate SHARP-Q under a consistent environment without supplementary corrections. For CNN architectures, to ensure a strictly fair comparison

with the current state-of-the-art CL-Calib, we adopt its exact identical configuration: maintaining the first and last layers at 8-bit and incorporating Distribution Correction (DC) (Liu et al., 2023). Finally, for Mamba super-resolution tasks, we utilize the DF2K (Lim et al., 2017) dataset for calibration, strictly adhering to standard protocols (Xin et al., 2026) to evaluate performance using PSNR and SSIM on the Y channel.

### 4.2. Comparison with State-of-the-Art Methods on ViTs

We evaluate SHARP-Q on representative Vision Transformer architectures against a comprehensive suite of state-of-the-art PTQ methods. Our comparison includes calibration-only approaches (e.g., PTQ4ViT (Yuan et al., 2022), RepQ-ViT (Li et al., 2023), AdaLog (Wu et al., 2024)) and reconstruction-based strategies (e.g., I&S-ViT (Zhong et al., 2023), DopQ-ViT (Yang et al., 2024), OASQ (Ma et al., 2024), FIMA-Q (Wu et al., 2025a), and APHQ-ViT (Wu et al., 2025b)).

As summarized in Table 1, SHARP-Q establishes new state-of-the-art performance across representative ViT architectures in both W4A4 and W3A3 settings. In the W4A4 regime, conventional uniform quantization frame-

*Table 3.* Ablation study of SHARP-Q components. We report Top-1 accuracy (%) on ImageNet. *MSE* denotes the standard reconstruction baseline.

| Components | | CNNs (W2A2) | | ViTs (W3A3) | |
|---|---|---|---|---|---|
| HAR | DFSC | ResNet-18 | RegX-600 | ViT-S | DeiT-T |
| - | - | 52.85 | 40.06 | 39.17 | 47.25 |
| ✓ | - | 54.18 | 41.69 | 40.51 | 48.63 |
| ✓ | ✓ | **56.74** | **46.89** | **66.93** | **56.14** |

works typically struggle to match specialized non-uniform baselines—such as AdaLog (Wu et al., 2024)—as the heavy-tailed activation distributions of ViTs often lead to severe geometric ill-conditioning. SHARP-Q, however, emerges as the only uniform quantization method to consistently outperform these specialized non-uniform benchmarks.

This performance advantage is further magnified in the even more aggressive W3A3 setting, where SHARP-Q consistently surpasses the nearest competitor by a substantial margin of **3.76%** on ViT-S. These results empirically validate that by accurately approximating the rectified Fisher Information Matrix, SHARP-Q enables hardware-friendly uniform quantizers to match or even exceed the representational capacity of specialized non-uniform designs. This suggests that under fixed bit-width constraints, the fidelity of quantization is primarily governed by the alignment with the intrinsic Fisher geometry, rather than the complexity of the quantizer design itself.

### 4.3. Comparison with State-of-the-Art Methods on CNNs

To demonstrate the architectural universality of SHARP-Q, we evaluate its performance on representative CNNs against a diverse array of baselines, including analytical methods (Banner et al., 2018; Nagel et al., 2020) and competitive reconstruction frameworks such as BRECQ (Li et al., 2021), QDrop (Wei et al., 2022), and CL-Calib (Shang et al., 2024).

As summarized in Table 2, prior methods encounter significant bottlenecks in the ultra-low-bit W2A2 regime: MSE-based methods (e.g., QDrop) treat all errors equally regardless of sensitivity, while block-diagonal proxies (e.g., BRECQ) neglect critical cross-dimensional couplings. Consequently, both strategies suffer from accuracy degradation in such constricted optimization spaces. By accurately approximating the rectified Fisher Information Matrix, SHARP-Q exhibits exceptional robustness and achieves gains over the previous state-of-the-art, CL-Calib, improving Top-1 accuracy by **2.29%** on ResNet-18, **3.11%** on MobileNetV2, and **4.70%** on RegNetX-600MF. This confirms that SHARP-Q captures the intrinsic Fisher geometry required for high-fidelity reconstruction, pushing the performance boundary of ultra-low-bit quantization.

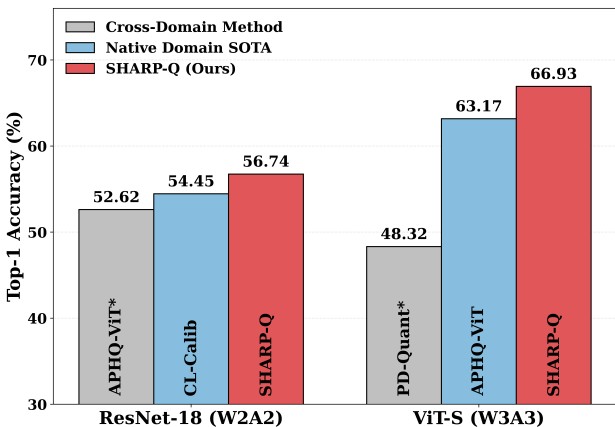

*Figure 3.* **Universality Analysis.** Performance comparison on ResNet-18 (W2A2) and ViT-S (W3A3). "Cross-Domain" denotes applying a method outside its native architecture (e.g., APHQ-ViT on CNNs). SHARP-Q demonstrates consistent superiority across both domains.

### 4.4. Ablation Study

To validate the contribution of individual components in SHARP-Q, we conduct ablation studies on representative architectures: ResNet-18 and RegNetX-600MF (CNNs) under W2A2, and ViT-S and DeiT-T (ViTs) under W3A3. As summarized in Table 3, we employ the standard MSE reconstruction as the baseline.

**Impact of HAR.** Incorporating Hessian-Aware Rectification (HAR) improves accuracy across all architectures. By explicitly preconditioning the optimization landscape, HAR compresses the global spectral envelope, thereby reducing interference from cross-dimensional couplings that isotropic MSE baseline fails to address. For instance, HAR alone improves accuracy by 1.63% on RegNetX-600MF and 1.34% on ViT-S. This confirms that alleviating ill-conditioning is a prerequisite for effective quantization, establishing a geometrically well-conditioned foundation for the subsequent high-fidelity recovery.

**Impact of DFSC.** Building on the rectified landscape, Dynamic Fisher-Subspace Compensation (DFSC) functions as the decisive step for high-precision reconstruction. By accurately approximating the rectified FIM via a hybrid surrogate, DFSC captures critical sensitivities that the isotropic MSE baseline inherently overlooks. This capability translates into substantial performance gains across all tested architectures; most notably on ViT-S, the combined strategy yields a **27.76%** improvement over the baseline. These results demonstrate that our "Rectify-then-Approximate" strategy enables the optimization to align with the intrinsic Fisher geometry, making high-fidelity recovery feasible even in ultra-low-bit regimes. (For a finer-grained ablation regarding the synergy between the static and dynamic terms within DFSC, please refer to Appendix A.1.)

## 4.5. Universality Analysis

To validate the architectural universality of SHARP-Q, we conduct a cross-domain evaluation by deploying domain-specific SOTA methods on non-native architectures (i.e., adapting APHQ-ViT for CNNs and PD-Quant for ViTs). We adhere to the standard configurations of the target domain, such as incorporating Distribution Correction (DC) for CNN evaluations. As illustrated in Figure 3, methods relying on specific heuristics exhibit marked degradation when crossing architectural boundaries. In contrast, SHARP-Q maintains high performance in both domains, surpassing even native baselines. This empirical success confirms that SHARP-Q possesses robust generalization capabilities, effectively transcending the limitations inherent in specialized heuristics and establishing a new standard for universal post-training quantization.

## 4.6. Extension to State Space Models

To validate that SHARP-Q's geometric principles transcend attention-based and convolutional paradigms, we extend our evaluation to State Space Models (SSMs). Specifically, we assess SHARP-Q on MambaIRv2-light (Guo et al., 2025) for image super-resolution tasks (W4A4), benchmarking against recent Mamba-specialized PTQ methods, including MambaQuant (Xu et al., 2025), Quamba (Chiang et al., 2025), and SPR$^2$Q (Xin et al., 2026).

Despite lacking architecture-specific heuristics, SHARP-Q consistently surpasses all specialized baselines across both $\times 2$ and $\times 4$ upscaling factors. For instance, on the Manga109 benchmark under the highly aggressive $\times 4$ resolution setting, SHARP-Q achieves a PSNR of 30.85 dB and an SSIM of 0.9091, outperforming the latest state-of-the-art SPR$^2$Q by a notable margin of **1.25 dB** in PSNR and **0.0132** in SSIM, while closely recovering the FP32 baseline (31.24 dB / 0.9182). Similar consistent performance gains are observed across the Set5, Set14, B100, and Urban100 datasets. The comprehensive quantitative tables are provided in Appendix A.5.

## 4.7. Further Analysis

To further demonstrate the versatility and robustness of SHARP-Q, we conduct extensive supplementary evaluations in the Appendix. Specifically, Appendix A.4 confirms our method's robust scalability to large-scale architectures (ViT-L, Swin-L) and superiority over current LLM-centric quantization approaches. Furthermore, Appendix A.3 highlights SHARP-Q's extreme data efficiency, where it outperforms conventional MSE baselines while requiring $16\times$ fewer calibration samples. Finally, detailed ablations on the subspace rank and offline reconstruction efficiency are provided in Appendices A.2 and A.6, respectively.

## 5. Conclusion

In this paper, we presented SHARP-Q, a unified post-training quantization framework grounded in Information Geometry. By aligning quantization with the intrinsic Fisher geometry rather than isotropic objectives, SHARP-Q implements a "Rectify-then-Approximate" strategy that rectifies optimization ill-conditioning to facilitate precise Fisher approximation for high-fidelity recovery. Extensive evaluations across diverse architectures confirm that SHARP-Q establishes new state-of-the-art performance. Crucially, our results reveal a pivotal insight: quantization fidelity is governed primarily by geometric alignment rather than quantizer complexity. This principle enables hardware-friendly uniform quantizers to outperform specialized non-uniform designs in ultra-low-bit regimes, translating geometric insights into tangible efficiency gains. Ultimately, this geometry-centric view highlights a critical transition in model compression strategies: extreme quantization depends less on quantizer sophistication and more on faithful alignment with intrinsic optimization geometry—a principle that grows increasingly fundamental as models scale toward foundation-level complexity.

## Acknowledgements

This work is supported by Zhejiang Province "Pioneering Soldier" and "Leading Goose" R&D Project (2023C01027).

## Impact Statement

This paper presents work whose goal is to advance the field of Machine Learning. There are many potential societal consequences of our work, none which we feel must be specifically highlighted here.

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

# A. Additional Experimental Results

## A.1. Hyperparameter Analysis: Synergy and Sensitivity

In this section, we conduct a comprehensive analysis of the hybrid objective in SHARP-Q. This analysis serves a dual purpose:

1. **Validating Component Synergy:** It provides a fine-grained ablation (supplementing Sec. 4.4) to demonstrate why coupling the static and dynamic terms is superior to using either alone.

2. **Examining Robustness:** It investigates the sensitivity to the hyperparameter $\alpha$ (supplementing the discussion in Sec. 4.5), verifying that SHARP-Q remains robust across a wide range of settings.

**Implementation Note.** In the main paper, the hybrid surrogate is formulated as $\mathbf{F}^{(t)}_{\text{hybrid}} = \mathbf{D}_{\text{static}} + \gamma \cdot \mathbf{S}^{(t)}_{\text{dynamic}}$ to emphasize the compensation role of the dynamic term. For numerical stability in our codebase, we implement this by applying the scaling factor to the static term: $\mathbf{F}'^{(t)}_{\text{hybrid}} = \alpha \cdot \mathbf{D}_{\text{static}} + \mathbf{S}^{(t)}_{\text{dynamic}}$. Here, $\alpha$ represents the effective weight ratio (mathematically equivalent to $1/\gamma$ in terms of relative importance).

**Results and Analysis.** We evaluate the Top-1 accuracy of DeiT-T and ViT-S under the challenging W3A3 setting by varying $\alpha$ from 0 (purely dynamic baseline) to 20. As illustrated in Figure 4, the performance follows a distinct "inverted-U" trend. This empirical evidence supports two key observations:

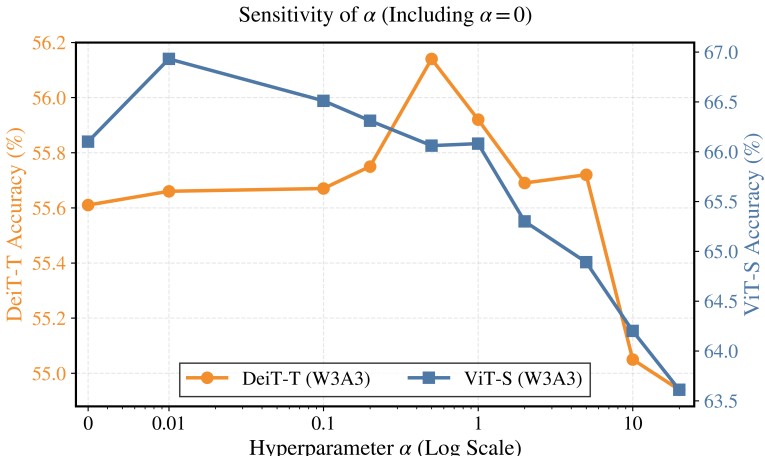

*Figure 4.* **Hyperparameter Sensitivity** ($\alpha$). Top-1 accuracy of DeiT-T and ViT-S (W3A3) across varying scaling factors $\alpha$. The observed "inverted-U" trajectory empirically validates the hybrid synergy: the optimal balance consistently outperforms both the purely dynamic baseline ($\alpha = 0$) and static-dominated regimes, while the broad peak indicates robustness to parameter selection.

- **Validation of Component Synergy (Ablation):** The necessity of the hybrid design is substantiated by the fact that the optimal hybrid configuration outperforms both extremes. When relying solely on the dynamic subspace term ($\alpha = 0$), the model achieves 66.10% on ViT-S. Introducing the static diagonal term ($\alpha = 0.01$) provides essential **full-rank stability**, boosting accuracy to **66.93%**. Conversely, when the static anchor dominates ($\alpha = 20$), the accuracy suffers a significant decline to 63.61% (-3.32% compared to the peak). This "inverted-U" behavior confirms that SHARP-Q requires the synergy of both components: the static term maintains global conditioning, while the dynamic term compensates for critical off-diagonal sensitivities that the static anchor misses.

- **Parameter Robustness (Sensitivity):** The sensitivity profile demonstrates that SHARP-Q is robust to hyperparameter variations within a reasonable range. Both models maintain high performance across a wide logarithmic scale (e.g., $\alpha \in [0.01, 1.0]$), alleviating the need for intensive per-model tuning. Regarding the optimal operating point, we observe architecture-dependent characteristics: ViT-S peaks at a smaller value ($\alpha = 0.01$), whereas DeiT-T prefers a more balanced ratio ($\alpha = 0.5$). Performance only degrades noticeably when $\alpha$ becomes excessively large (e.g., $\alpha = 20$), as the static term begins to overshadow the critical dynamic compensation.

To further underscore the method's universality, we adopt unified configurations for other hyperparameters. Subspace Rank ($r$): We fix $r = 5$ across all reported experiments, eliminating the need for architecture-specific rank tuning and confirming that once the landscape is rectified by HAR, a low-rank approximation suffices to capture dominant curvature. Trust Region ($\lambda$): We empirically adopt $\lambda = 2.0$ as a robust default setting. While minor architecture-specific tuning could yield marginal gains, this standard configuration consistently achieves high-fidelity rectification across diverse architectures without requiring intensive search.

### A.2. Ablation on Subspace Rank ($r$)

To further validate our hyperparameter selection, we provide a detailed ablation on the subspace rank $r$ across CNN (ResNet-18, RegX-600M under W2A2) and ViT (ViT-S, DeiT-T under W3A3) architectures. As shown in Table 4, increasing the rank beyond $r = 5$ yields marginal performance gains but introduces a disproportionate increase in calibration time. The default setting of $r = 5$ strikes an optimal balance, effectively capturing the dominant curvature while maintaining high computational efficiency.

*Table 4.* Ablation on subspace rank $r$ across CNN (W2A2) and ViT (W3A3) architectures.

| Rank ($r$) | ResNet-18 | RegX-600M | ViT-S | DeiT-T | Calib. Time (ViT-S) |
|---|---|---|---|---|---|
| 1 | 56.12% | 45.80% | 65.39% | 55.36% | 1.77h |
| **5 (Default)** | **56.74%** | **46.89%** | **66.93%** | **56.14%** | **1.96h** |
| 10 | 56.81% | 47.15% | 67.12% | 56.03% | 2.65h |
| 20 | 56.62% | 47.29% | 66.75% | 56.29% | 4.12h |

### A.3. High Data-Efficiency with Calibration Data

To assess data-dependency, we ablated the calibration data size on ViT-S under the W3A3 setting. As summarized in Table 5, SHARP-Q exhibits exceptional robustness and data-efficiency compared to the MSE baseline. This confirms that our surrogate matrix captures generalized properties of the loss landscape rather than overfitting to specific calibration samples, ensuring high robustness for practical deployment even with severely limited data.

*Table 5.* Ablation on calibration data size for ViT-S (W3A3).

| Data Size | 128 | 256 | 512 | 1024 (standard) | 2048 |
|---|---|---|---|---|---|
| MSE | 13.48% | 30.15% | 37.36% | 39.17% | 45.98% |
| SHARP-Q (Ours) | 52.76% | 58.98% | 63.58% | 66.93% | 68.59% |

### A.4. Scalability and Practical Limitation

**Scalability to Large Architectures.** We confirm that SHARP-Q scales robustly to large-scale architectures. With a fixed $r = 5$, it captures dominant curvature effectively even as dimensionality increases, as evidenced by our ViT-L and Swin-L results on a single RTX A6000 (Table 6).

*Table 6.* Scalability evaluation on large-scale Vision Transformers.

| Model | FP32 | W4A4 | W3A3 |
|---|---|---|---|
| ViT-L | 85.86% | 84.81% | 80.53% |
| Swin-L | 86.30% | 85.04% | 80.77% |

**Comparison with LLM-centric Methods.** To provide a comprehensive analysis, we benchmark against prominent LLM-centric post-training quantization methods, including GPTQ (Frantar et al., 2022) and the recently proposed GPTAQ

(Li et al., 2025). As shown in Table 7, methods designed specifically for LLMs often generalize poorly to distinct visual architectures, showing significant accuracy degradation. In contrast, SHARP-Q maintains superior representational fidelity on these vision models.

Table 7. Comparison with LLM-centric methods (W4A4).

| Model (W4A4) | FP32 | GPTQ | GPTAQ | SHARP-Q (Ours) |
|---|---|---|---|---|
| DeiT-S | 79.85% | 71.9% | 72.8% | **76.86%** |
| DeiT-B | 81.80% | 77.7% | 78.4% | **80.50%** |

**Limitation (LLMs).** Extending SHARP-Q to 7B+ LLMs is currently constrained by the iterative optimization overhead of AdaRound-based reconstruction. Bridging this gap via more efficient solvers is identified as key future work.

### A.5. Universality beyond CNN and ViT: State Space Models (Mamba)

To validate that SHARP-Q's geometric principles transcend specific architectures, we further evaluated it on the Mamba architecture for Image Super-Resolution.

**Experimental Setup.** Following the standard evaluation protocol in Mamba quantization literature (Xin et al., 2026), we adopt the full-precision MambaIRv2-light (Guo et al., 2025) as our base model for $\times 2$ and $\times 4$ super-resolution tasks. For calibration, we strictly adhere to the calibration protocols established in the baseline settings to ensure fairness. Performance is evaluated on five standard benchmark datasets: Set5 (Bevilacqua et al., 2012), Set14 (Zeyde et al., 2010), B100 (Martin et al., 2001), Urban100 (Huang et al., 2015), and Manga109 (Matsui et al., 2017). Following conventional SR evaluation metrics, Peak Signal-to-Noise Ratio (PSNR) and Structural Similarity Index (SSIM) are calculated on the Y channel of the YCbCr space.

**Baseline Comparison.** We benchmark SHARP-Q against recent state-of-the-art PTQ methods specifically tailored for state-space models, including MambaQuant (Xu et al., 2025), Quamba (Chiang et al., 2025), 2Dquant (Liu et al., 2024), and SPR$^2$Q (Xin et al., 2026). To ensure a strictly fair and rigorous comparison, all baseline methods are evaluated under a unified framework on the exact same MambaIRv2-light backbone.

Notably, SHARP-Q outperforms the latest Mamba-specialized SOTA. To our knowledge, SHARP-Q is the first PTQ method to achieve consistent SOTA across CNN, ViT, and Mamba paradigms simultaneously, confirming its exceptional universality and ability to deliver superior performance regardless of model structure.

Table 8. 4-bit (W4A4) Quantization on MambaIRv2-light (PSNR / SSIM).

| Method | Scale | Set5 | Set14 | B100 | Urban100 | Manga109 |
|---|---|---|---|---|---|---|
| MambaIRv2 (FP32) | $\times 2$ | 38.26/0.9615 | 34.07/0.9221 | 32.36/0.9019 | 33.26/0.9378 | 39.35/0.9785 |
| MambaQuant (Xu et al., 2025) | $\times 2$ | 36.67/0.9495 | 31.76/0.8899 | 30.85/0.8756 | 28.08/0.8407 | 33.47/0.9186 |
| Quamba (Chiang et al., 2025) | $\times 2$ | 37.07/0.9544 | 32.77/0.9092 | 31.47/0.8896 | 30.54/0.9107 | 36.94/0.9699 |
| 2Dquant (Liu et al., 2024) | $\times 2$ | 37.34/0.9560 | 33.01/0.9123 | 31.66/0.8923 | 30.79/0.9141 | 37.35/0.9718 |
| SPR$^2$Q (Xin et al., 2026) | $\times 2$ | 37.72/0.9589 | 33.27/0.9156 | 31.94/0.8966 | 31.53/0.9223 | 38.03/0.9754 |
| **SHARP-Q (Ours)** | $\times 2$ | **38.05/0.9612** | **33.64/0.9192** | **32.11/0.8996** | **31.95/0.9287** | **38.91/0.9788** |
| MambaIRv2 (FP32) | $\times 4$ | 32.51/0.8992 | 28.84/0.7877 | 27.75/0.7426 | 26.82/0.8079 | 31.24/0.9182 |
| MambaQuant (Xu et al., 2025) | $\times 4$ | 30.74/0.8650 | 27.17/0.7413 | 26.37/0.6920 | 23.28/0.6694 | 26.73/0.8186 |
| Quamba (Chiang et al., 2025) | $\times 4$ | 31.01/0.8715 | 27.77/0.7585 | 26.99/0.7149 | 25.01/0.7470 | 28.57/0.8752 |
| 2Dquant (Liu et al., 2024) | $\times 4$ | 31.28/0.8774 | 27.99/0.7644 | 27.14/0.7201 | 25.30/0.7573 | 29.05/0.8851 |
| SPR$^2$Q (Xin et al., 2026) | $\times 4$ | 31.60/0.8844 | 28.27/0.7725 | 27.33/0.7274 | 25.64/0.7713 | 29.60/0.8959 |
| **SHARP-Q (Ours)** | $\times 4$ | **32.13/0.8924** | **28.57/0.7811** | **27.59/0.7348** | **26.06/0.7844** | **30.85/0.9091** |

## A.6. Training Efficiency Analysis

We evaluate the computational efficiency of SHARP-Q using a single NVIDIA A6000 GPU, with results summarized in Table 9.

**Clarification on Time Metric.** It is important to clarify that the efficiency metric discussed in this section refers exclusively to the **offline reconstruction cost**—i.e., the time required to convert a pre-trained floating-point model into a quantized one using the calibration set. This is a one-time optimization process and does not affect the inference latency during deployment.

**Computational Complexity Perspective.** Although SHARP-Q is grounded in the **intrinsic Fisher geometry**, it avoids the prohibitive $\mathcal{O}(N^2)$ complexity of explicit Hessian construction. Both HAR and DFSC operate via our Fisher-KL Oracle, which corresponds to the cost of standard backpropagation. The dynamic subspace correction maintains only $r$ historical perturbation vectors ($r = 5$ in all experiments), leading to linear space-time complexity:

- **Computational Cost:** $\mathcal{O}(Nr)$ per iteration;
- **Memory Overhead:** $\mathcal{O}(Nr)$,

where $N$ denotes the dimensionality of the block output (i.e., channel or token dimension).

Since we utilize a compact subspace rank ($r = 5$), this additional memory footprint is marginal. In our experiments, the peak memory usage is comfortably accommodated by standard GPUs (e.g., NVIDIA RTX A6000), ensuring practical feasibility without requiring specialized high-memory clusters.

*Table 9.* Comparison of reconstruction efficiency and accuracy on ImageNet. All methods are evaluated on a single NVIDIA A6000 GPU using a small calibration set. Note that "Recon. Time" denotes the one-time cost for quantizing the model.

| Model | Method | Data Size | Recon. Time (h) | Acc. (%) |
|---|---|---|---|---|
| **ResNet-18** (W2A2) | PD-Quant (Liu et al., 2023) | 1024 | 1.11 | 53.08 |
| | CL-Calib (Shang et al., 2024) | 1024 | >2.00 | 54.45 |
| | **SHARP-Q (Ours)** | 1024 | 1.37 | **56.74** |
| **ViT-S** (W3A3) | APHQ-ViT (Wu et al., 2025b) | 1024 | 1.55 | 63.17 |
| | FIMA-Q (Wu et al., 2025a) | 1024 | 1.72 | 62.67 |
| | **SHARP-Q (Ours)** | 1024 | 1.96 | **66.93** |

**Results and Analysis.** As shown in Table 9, SHARP-Q achieves a favorable efficiency–accuracy trade-off among PTQ methods.

On ResNet-18, SHARP-Q secures a remarkable **3.66%** absolute improvement over PD-Quant. This substantial performance leap is achieved with only a marginal increase in reconstruction time, representing a highly favorable accuracy-efficiency trade-off. Particularly compelling is the comparison against the recent state-of-the-art CL-Calib: SHARP-Q not only outperforms it by **2.29%** in accuracy but also achieves this with significantly faster reconstruction speed, effectively eliminating the heavy computational burden associated with contrastive learning. This dominance extends to ViT-S, where SHARP-Q establishes a new accuracy benchmark of **66.93%**. It decisively surpasses FIMA-Q, achieving superior fidelity within a highly competitive reconstruction budget, thereby setting a new standard for the accuracy-efficiency trade-off.

Crucially, SHARP-Q remains orders of magnitude faster than Quantization-Aware Training (QAT). For instance, LSQ (Esser et al., 2019) typically requires approximately 96 hours to fine-tune ResNet-18 on the full ImageNet dataset ($\sim$1.28 million images). In sharp contrast, SHARP-Q completes the process in only 1.37 hours—approximately **70×** faster—using a tiny calibration set (1,024 images). This confirms that SHARP-Q successfully bridges the accuracy gap between PTQ and QAT while preserving the fast deployment capability essential for real-world applications.

# B. Proof of Proposition 1: Monotonic Spectral Compression

**Objective.** In Proposition 3.1, we assert that the HAR update rule monotonically minimizes the rectified curvature energy. This derivation rigorously establishes that minimizing the trace of this energy creates a well-conditioned landscape by implicitly compressing the spectral envelope and suppressing cross-dimensional couplings.

**1. Formulation of the Rectified Objective.** Based on the Fisher-KL relationship (Eq. 11), the local degradation of the optimization objective corresponds to the Fisher-weighted quadratic form:

$$\mathcal{L}_{\text{KL}} \approx \frac{1}{2}\Delta \mathbf{z}^\top \mathbf{F} \Delta \mathbf{z}. \tag{24}$$

In the rectification stage, the linear layer is reparameterized as $\tilde{\mathbf{Y}} = (\mathbf{W}\boldsymbol{\Gamma})(\boldsymbol{\Gamma}^{-1}\mathbf{X})$. We model quantization as an additive noise vector $\boldsymbol{\epsilon}$ injected into the rectified input space $\tilde{\mathbf{X}} = \boldsymbol{\Gamma}^{-1}\mathbf{X}$. This noise propagates through the fused weights $\tilde{\mathbf{W}} = \mathbf{W}\boldsymbol{\Gamma}$, resulting in the output perturbation:

$$\Delta \mathbf{z} = \tilde{\mathbf{W}}\boldsymbol{\epsilon} = \mathbf{W}\boldsymbol{\Gamma}\boldsymbol{\epsilon}. \tag{25}$$

**2. Derivation of the Trace Objective.** We aim to minimize the expected curvature energy $\mathcal{J}(\mathbf{P}) \triangleq \mathbb{E}_{\boldsymbol{\epsilon}}[\mathcal{L}_{\text{KL}}]$. Adopting the standard PTQ assumption that quantization noise $\boldsymbol{\epsilon}$ is zero-mean and uncorrelated with uniform variance $\sigma^2$ (i.e., $\mathbb{E}[\boldsymbol{\epsilon}\boldsymbol{\epsilon}^\top] = \sigma^2 \mathbf{I}$), we derive the explicit trace form.

First, substituting $\Delta \mathbf{z}$ into the quadratic form:

$$\mathbb{E}[\mathcal{L}_{\text{KL}}] \approx \frac{1}{2}\mathbb{E}_{\boldsymbol{\epsilon}}\left[(\mathbf{W}\boldsymbol{\Gamma}\boldsymbol{\epsilon})^\top \mathbf{F}(\mathbf{W}\boldsymbol{\Gamma}\boldsymbol{\epsilon})\right] = \frac{1}{2}\mathbb{E}_{\boldsymbol{\epsilon}}\left[\boldsymbol{\epsilon}^\top (\boldsymbol{\Gamma}^\top \mathbf{W}^\top \mathbf{F}\mathbf{W}\boldsymbol{\Gamma})\boldsymbol{\epsilon}\right]. \tag{26}$$

Let $\mathbf{M} \triangleq \boldsymbol{\Gamma}\mathbf{W}^\top \mathbf{F}\mathbf{W}\boldsymbol{\Gamma}$ be the rectified curvature matrix projected onto the input space (note that $\boldsymbol{\Gamma}$ is diagonal and symmetric, so $\boldsymbol{\Gamma}^\top = \boldsymbol{\Gamma}$). Using the trace cyclic property $\mathbf{x}^\top \mathbf{A}\mathbf{x} = \text{Tr}(\mathbf{A}\mathbf{x}\mathbf{x}^\top)$, we can move the expectation inside the trace:

$$\begin{aligned}
\mathcal{J}(\mathbf{P}) &= \frac{1}{2}\mathbb{E}_{\boldsymbol{\epsilon}}\left[\text{Tr}\left(\mathbf{M}\boldsymbol{\epsilon}\boldsymbol{\epsilon}^\top\right)\right] \\
&= \frac{1}{2}\text{Tr}\left(\mathbf{M} \cdot \mathbb{E}[\boldsymbol{\epsilon}\boldsymbol{\epsilon}^\top]\right) \\
&= \frac{1}{2}\text{Tr}\left(\mathbf{M} \cdot \sigma^2 \mathbf{I}\right) = \frac{\sigma^2}{2}\text{Tr}(\mathbf{M}).
\end{aligned} \tag{27}$$

Substituting $\mathbf{M}$ back, and defining the parameter-space curvature as $\hat{\mathbf{F}} \triangleq \mathbf{W}^\top \mathbf{F}\mathbf{W}$:

$$\mathcal{J}(\mathbf{P}) = \frac{\sigma^2}{2}\text{Tr}\left(\boldsymbol{\Gamma}\hat{\mathbf{F}}\boldsymbol{\Gamma}\right). \tag{28}$$

Since $\boldsymbol{\Gamma} = \text{diag}(\gamma_1, \ldots, \gamma_C)$ is a diagonal matrix, the diagonal elements of the product $\boldsymbol{\Gamma}\hat{\mathbf{F}}\boldsymbol{\Gamma}$ are simply $(\boldsymbol{\Gamma}\hat{\mathbf{F}}\boldsymbol{\Gamma})_{ii} = \gamma_i \hat{F}_{ii}\gamma_i = \gamma_i^2 \hat{F}_{ii}$. Thus, the final scalar objective is:

$$\mathcal{J}(\mathbf{P}) \propto \sum_{i=1}^{C_{\text{in}}} \gamma_i^2 \hat{F}_{ii} = \sum_{i=1}^{C_{\text{in}}} e^{2p_i} \hat{F}_{ii}. \tag{29}$$

This confirms that minimizing the KL divergence is equivalent to minimizing the weighted trace of the curvature matrix.

**3. Monotonic Descent Dynamics.** The HAR update rule (Eq. 18) corresponds to a standard gradient descent step on the objective $\mathcal{J}(\mathbf{P})$ (augmented with a strictly convex trust-region term). Since $\mathcal{J}(\mathbf{P})$ is differentiable with respect to $\mathbf{P}$, the *Descent Lemma* guarantees that for a sufficiently small step size $\eta \leq 1/L$ (where $L$ is the Lipschitz constant of $\nabla \mathcal{J}$), the objective value decreases monotonically:

$$\mathcal{J}(\mathbf{P}_{t+1}) \leq \mathcal{J}(\mathbf{P}_t) - \frac{\eta}{2}\|\nabla_{\mathbf{P}}\mathcal{J}\|_2^2. \tag{30}$$

This mathematically proves the **monotonic reduction** of the rectified curvature energy trace asserted in Proposition 3.1.

**4. Mechanism of Spectral Compression.** We now prove that minimizing this trace effectively compresses the spectral envelope (eigenvalues) and suppresses off-diagonal couplings. Let $\tilde{\mathbf{F}} = \boldsymbol{\Gamma}\hat{\mathbf{F}}\boldsymbol{\Gamma}$ be the rectified curvature matrix with elements $\tilde{F}_{ij}$.

- **Eigenvalue Sum Minimization:** The trace of a matrix is invariant and equals the sum of its eigenvalues: $\mathrm{Tr}(\tilde{\mathbf{F}}) = \sum_k \lambda_k(\tilde{\mathbf{F}})$. Since $\hat{\mathbf{F}}$ is Positive Semi-Definite (PSD), $\tilde{\mathbf{F}}$ remains PSD, ensuring $\lambda_k \geq 0$. Therefore, monotonically reducing the trace $\mathcal{J}(\mathbf{P})$ directly minimizes the sum of the eigenvalues, thereby compressing the global **spectral envelope** of the optimization landscape.

- **Off-Diagonal Bounding:** By the Cauchy-Schwarz inequality for PSD matrices, the off-diagonal elements are bounded by the diagonal entries: $|\tilde{F}_{ij}|^2 \leq \tilde{F}_{ii}\tilde{F}_{jj}$. As the optimization reduces the diagonal energy terms $\tilde{F}_{ii}$ (via scaling factors $\gamma_i$), the upper bounds for the cross-dimensional couplings $|\tilde{F}_{ij}|$ are simultaneously tightened. This theoretical guarantee confirms that HAR transforms the ill-conditioned surface into a more isotropic geometry, validating the claims in Proposition 3.1.

$\square$

## C. Proof of Proposition 2: Derivation and Spectral Analysis

**Objective.** In Proposition 3.2, we assert that the dynamic subspace correction acts as a spectrally consistent proxy for the true Fisher geometry. Here, we provide a rigorous derivation of the quadratic form induced by DFSC and analyze its spectral properties.

**1. Algebraic Derivation of the Surrogate Quadratic Form.** Recall the definition of the dynamic correction term in our reconstruction objective:

$$\mathcal{Q}_{\mathrm{corr}} \triangleq \gamma \cdot \Delta\mathbf{z}^\top \mathbf{S}_{\mathrm{dynamic}}^{(t)} \Delta\mathbf{z}. \tag{31}$$

Substituting the definition of $\mathbf{S}_{\mathrm{dynamic}}^{(t)}$ from Eq. (21), we expand the quadratic form as:

$$\mathcal{Q}_{\mathrm{corr}} = \gamma \cdot \Delta\mathbf{z}^\top \left[ \mathbf{G}_{\mathrm{mem}}(\Delta\mathbf{Z}_{\mathrm{mem}}^\top \Delta\mathbf{Z}_{\mathrm{mem}})^{-1} \mathbf{G}_{\mathrm{mem}}^\top \right] \Delta\mathbf{z}. \tag{32}$$

*Remark on Stability:* We empirically observe that the correlation matrix $\mathbf{C}_{\mathrm{sub}} = \Delta\mathbf{Z}_{\mathrm{mem}}^\top \Delta\mathbf{Z}_{\mathrm{mem}}$ remains well-conditioned.[1]

To analyze the geometry, we invoke the **Fisher-KL Oracle** relationship: $\mathbf{G}_{\mathrm{mem}} \approx \mathbf{F}\Delta\mathbf{Z}_{\mathrm{mem}}$ (Eq. 20). Substituting this into Eq. (32) yields:

$$\mathcal{Q}_{\mathrm{corr}} \approx \gamma \cdot \Delta\mathbf{z}^\top \mathbf{F} \underbrace{\left[ \Delta\mathbf{Z}_{\mathrm{mem}}(\Delta\mathbf{Z}_{\mathrm{mem}}^\top \Delta\mathbf{Z}_{\mathrm{mem}})^{-1} \Delta\mathbf{Z}_{\mathrm{mem}}^\top \right]}_{\mathbf{P}_{\mathcal{S}}} \mathbf{F}\Delta\mathbf{z}. \tag{33}$$

The bracketed term $\mathbf{P}_{\mathcal{S}}$ is the *orthogonal projection operator* onto the subspace $\mathcal{S}$ spanned by the columns of $\Delta\mathbf{Z}_{\mathrm{mem}}$.

**2. Subspace Invariance and Gradient Matching.** For a perturbation $\Delta\mathbf{z}$ lying within the captured error subspace $\mathcal{S}_{\mathrm{err}}^{(t)}$, we invoke the *dominant curvature assumption*. Since $\mathcal{S}$ is constructed from dynamic perturbations collected during the optimization trajectory, it spans the directions where the loss is most sensitive to noise. In such an aligned coordinate system, the vector $\mathbf{F}\Delta\mathbf{z}$ (representing the local curvature response) resides predominantly within $\mathcal{S}$. Consequently, the projection operator $\mathbf{P}_{\mathcal{S}}$ acts as an approximate identity mapping for these principal directions, yielding:

$$\mathcal{Q}_{\mathrm{corr}} \approx \gamma \cdot (\mathbf{F}\Delta\mathbf{z})^\top (\mathbf{F}\Delta\mathbf{z}) = \gamma \cdot \|\mathbf{F}\Delta\mathbf{z}\|_2^2. \tag{34}$$

**Theoretical Interpretation.** Equation (34) reveals that DFSC minimizes the squared norm of the Fisher-vector product. Since $\mathbf{F}\Delta\mathbf{z} \approx \nabla_{\mathbf{z}}\mathcal{L}$, this confirms that our objective is mathematically equivalent to **Gradient Matching** in the output space:

$$\min_{\Theta_q} \|\mathbf{F}\Delta\mathbf{z}\|_2^2 \iff \min_{\Theta_q} \|\nabla_{\mathbf{z}}\mathcal{L}(\mathbf{z} + \Delta\mathbf{z}) - \nabla_{\mathbf{z}}\mathcal{L}(\mathbf{z})\|_2^2. \tag{35}$$

This drives the quantized model towards a stationary point of the optimization landscape, providing a stronger optimization signal than simple loss value matching.

**3. Spectral Properties.** Let $\mathbf{F} = \mathbf{V}\mathbf{\Lambda}\mathbf{V}^\top$ be the eigendecomposition of the Fisher matrix. The surrogate operator $\mathbf{F}^2 = \mathbf{V}\mathbf{\Lambda}^2\mathbf{V}^\top$ possesses the following properties:

---

[1]Theoretically, this is attributable to the high dimensionality of the activation space relative to the subspace rank ($N \gg r$), which renders fortuitous linear dependence statistically negligible.

- **Eigenspace Alignment:** The surrogate shares an identical set of eigenvectors with the true Fisher matrix. This formalizes the assertion in Proposition 3.2 that DFSC's dynamic correction term aligns its penalty with the principal directions of the optimization landscape.

- **Order Preservation:** The eigenvalues transform via $\lambda_i \to \lambda_i^2$. Since $f(x) = x^2$ is strictly monotonic for $x \geq 0$, the relative priority of curvature directions is preserved ($\lambda_i > \lambda_j \implies \lambda_i^2 > \lambda_j^2$).

- **Curvature Sharpening:** The quadratic scaling imposes a super-linear penalty on dominant eigenvalues. This ensures that quantization noise aligned with critical directions is suppressed more aggressively, which is essential for maintaining stability in ultra-low-bit regimes.

$\square$

## D. Deployment Efficiency and Computational Overhead

In the main text, we state that the inverse rectification term $\mathbf{\Gamma}^{-1}$ is handled via either "efficient operator fusion" or "lightweight pre-scaling." Here, we provide a granular analysis of these two mechanisms and rigorously derive the computational cost for both ResNet-18 and ViT-Small.

### D.1. Mechanism of Action

The rectified forward pass is defined as $\tilde{\mathbf{Y}} = (\mathbf{W}\mathbf{\Gamma})(\mathbf{\Gamma}^{-1}\mathbf{X})$. While the term $(\mathbf{W}\mathbf{\Gamma})$ is always fused offline into the weights, the handling of the input transformation $(\mathbf{\Gamma}^{-1}\mathbf{X})$ depends on the architectural context of the activation $\mathbf{X}$:

- **Case 1: Efficient Operator Fusion (Zero Overhead).** If $\mathbf{X}$ is the output of a normalization layer (e.g., LayerNorm or BatchNorm) with affine parameters $\boldsymbol{\gamma}_{\mathrm{ln}}, \boldsymbol{\beta}_{\mathrm{ln}}$, the inverse rectification $\mathbf{\Gamma}^{-1}$ can be mathematically merged into the normalization statistics offline.

$$\mathbf{\Gamma}^{-1} \cdot \mathrm{LN}(\mathbf{x}) = \mathbf{\Gamma}^{-1} \cdot \left( \frac{\mathbf{x} - \mu}{\sigma} \odot \boldsymbol{\gamma}_{\mathrm{ln}} + \boldsymbol{\beta}_{\mathrm{ln}} \right) = \frac{\mathbf{x} - \mu}{\sigma} \odot \underbrace{(\mathbf{\Gamma}^{-1}\boldsymbol{\gamma}_{\mathrm{ln}})}_{\boldsymbol{\gamma}'_{\mathrm{ln}}} + \underbrace{(\mathbf{\Gamma}^{-1}\boldsymbol{\beta}_{\mathrm{ln}})}_{\boldsymbol{\beta}'_{\mathrm{ln}}}. \tag{36}$$

This results in updated normalization parameters $\boldsymbol{\gamma}'_{\mathrm{ln}}, \boldsymbol{\beta}'_{\mathrm{ln}}$ with zero additional FLOPs at inference time.

- **Case 2: Lightweight Pre-scaling (Marginal Overhead).** If $\mathbf{X}$ originates from a non-linear activation (e.g., GELU, Softmax) or a residual connection without immediate normalization, fusion is not directly applicable. In this scenario, we apply explicit element-wise multiplication: $\mathbf{x}' = \mathbf{x} \odot \mathrm{diag}(\mathbf{\Gamma}^{-1})$. The cost is $N \cdot C_{\mathrm{in}}$ FLOPs.

### D.2. Granular Analysis on ViT-Small

We apply this logic to a standard Transformer Block (ViT-Small, sequence length $N$, hidden dimension $D$).

**1. Multi-Head Self-Attention (MSA) Module.** The MSA contains 4 linear projections: Query ($W_q$), Key ($W_k$), Value ($W_v$), and Output ($W_{\mathrm{proj}}$).

- **Inputs to $W_q, W_k, W_v$:** These projections immediately follow the block's first LayerNorm (LN1).
  - *Mechanism:* Efficient operator fusion. We merge $\mathbf{\Gamma}^{-1}$ into LN1.
  - *Cost:* 0 FLOPs.

- **Input to $W_{\mathbf{proj}}$:** This projection acts on the weighted sum of values (output of Softmax attention). No normalization exists here.
  - *Mechanism:* Lightweight pre-scaling.
  - *Cost:* $N \cdot D$ FLOPs.

**2. MLP Module.** The MLP contains 2 linear projections: Expansion ($W_{\mathrm{fc1}}$) and Contraction ($W_{\mathrm{fc2}}$, expansion ratio $E = 4$).

- **Input to $W_{\mathbf{fc1}}$:** This projection immediately follows the block's second LayerNorm (LN2).

- *Mechanism:* Efficient operator fusion. We merge $\Gamma^{-1}$ into LN2.
- *Cost:* 0 FLOPs.

- **Input to $W_{\mathbf{fc2}}$:** This projection follows the GELU activation function.

  - *Mechanism:* Lightweight pre-scaling.
  - *Cost:* $N \cdot (4D)$ FLOPs (since the hidden dimension expands to $4D$).

**Global Ratio Calculation (ViT).** Summing the overheads:

$$\mathcal{F}_{\text{overhead}} = \underbrace{(N \cdot D)}_{\text{MSA Proj}} + \underbrace{(N \cdot 4D)}_{\text{MLP FC2}} = 5ND \tag{37}$$

The total computational cost of a standard ViT block ($\mathcal{F}_{\text{model}}$) consists of the dense linear projections and the core attention matrix multiplications ($QK^\top$ and $AV$):

$$\mathcal{F}_{\text{model}} = \underbrace{12ND^2}_{\text{Linear Projections}} + \underbrace{2N^2D}_{\text{Attention Matrix Multiplies}} \tag{38}$$

For ViT-Small ($D = 384$) operating on the standard ImageNet resolution ($16 \times 16$ patches, sequence length $N = 197$), the exact overhead ratio is:

$$R_{\text{vit}} = \frac{5ND}{12ND^2 + 2N^2D} = \frac{5}{12D + 2N} = \frac{5}{12(384) + 2(197)} \approx \mathbf{0.100\%}. \tag{39}$$

### D.3. Global Analysis on ResNet-18

For Convolutional Networks, the standard block structure is Conv $\rightarrow$ BN $\rightarrow$ ReLU. The input to the next convolution typically comes from a ReLU activation.

- *Conservative Estimation:* Although $\Gamma$ is a positive diagonal matrix ($\exp(\mathbf{P})$) and could theoretically commute with ReLU (since $\text{ReLU}(c \cdot x) = c \cdot \text{ReLU}(x)$ for $c > 0$) to be fused into the preceding BN, we adopt a worst-case assumption for rigor. We assume lightweight pre-scaling is required for every convolutional layer.

Let $\mathcal{L}$ denote all convolutional layers. The global overhead ratio is bounded by the layer with the smallest channel width ($C_{\text{out}} = 64$) and kernel size ($K = 3$):

$$R_{\text{global}} = \frac{\sum HWC_{\text{in}}}{\sum HWC_{\text{in}}C_{\text{out}}K^2} \leq \max_{l} \left( \frac{1}{C_{\text{out}}^{(l)} \cdot K_l^2} \right). \tag{40}$$

For ResNet-18, this conservative upper bound yields $R_{\text{global}} \leq \frac{1}{64 \times 9} \approx \mathbf{0.17\%}$.

**Summary.** Our rigorous analysis confirms that SHARP-Q imposes a mathematically negligible computational burden. By leveraging efficient operator fusion, we effectively minimize the explicit rectification cost in ViT architectures. Even under conservative worst-case assumptions for CNNs, the total theoretical overhead remains strictly bounded below $\mathbf{0.2\%}$. Consequently, this minute cost is effectively invisible in real-world deployment, allowing SHARP-Q to match the inference latency of standard uniform quantization baselines while delivering significantly superior accuracy.

