# OpenReview forum: "SHARP-Q: Spectral Hessian Alignment and Rectification for Post-training Quantization"
_ICML.cc/2026/Conference — ICML 2026 regular_

### Official Review · Reviewer_ThMh · 2026-03-05

**Soundness:** 2
**Presentation:** 3
**Significance:** 3
**Originality:** 2
**Overall Recommendation:** 4
**Confidence:** 4

**Summary:**

This paper introduces SHARP-Q, a post-training quantization method for vision transformers and CNNs. At first, this paper says Mean Squared Error (MSE) implicitly assume parameter independence, making these target failing to capture the complex cross-dimensional couplings essential for performance recovery. Then, this paper also claim that some strategies (FIMA-Q and APHQ-ViT) incorporate curvature information; their effectiveness is constrained by the underlying landscape’s ill-conditioning. Thus, SHARP-Q first actively transforms the optimization geometry into a well-conditioned foundation through Hessian-aware rectification. It then applies Dynamic Fisher-Subspace Compensation to accurately approximates the Fisher information matrix. As the experiments show, this SHARP-Q achieves the best performance on many vision transformers and cnn in ultra-low bit cases.

**Compliance With Llm Reviewing Policy:**

Affirmed.

**Final Justification:**

My concerns are solved. I decide to raise my rating. Please carefully revise the paper, and I sincerely hope that you can open-source your code

**Key Questions For Authors:**

The motivation should be clarified.

**Limitations:**

The author seems not discussed the limitations and potential negative societal impact of their work.

**Strengths And Weaknesses:**

Strengths

1.	Clear illustrations.

2.	Good performance.

Weaknesses:

1.	As the paper said, the previous methods (FIMA-Q and APHQ-ViT) encountered an ill-conditioning problem. How does this paper define this problem? This paper claims to solve this problem, but what is the evidence? I look forward to seeing specific instructions to support these claims.

2.	Hessian-Aware Rectification is actually very similar to OmniQuant's method.

3.	According to Figure 2, Hessian-Aware Rectification seems to use the output of the entire network for rectification, which seems to introduce a huge overhead for larger models.

4.	There seems to be no description on how to initialize P_{init} in the paper.

5.	Line 242, why should this paper avoid the computational overhead of explicit spectral decomposition? I don't understand why spectral decomposition is needed.

6.	Line 311, why HAR stage have a subspace rank of r = 5? Is rank r only needed in Stage 2: Dynamic Fisher-Subspace Compensation?

---

> ### Author Rebuttal · Authors · 2026-03-28
>
> We thank the reviewer for the meticulous feedback and for recognizing our strong performance. We highly appreciate the attention to technical detail and provide point-by-point clarifications below to address each concern.
> ***
> **Q1: Motivation and Universal Generalization (KQ1, Motivation)**
>
> Our motivation is to establish a universal geometric principle for PTQ. Beyond the CNN and ViT results in our paper, we further validated SHARP-Q on Mamba (MambaIRv2); it remarkably outperforms the latest Mamba-specific SOTA, SPR$^2$Q (ICLR 2026). To our knowledge, **SHARP-Q is the first PTQ method to achieve consistent SOTA across CNN, ViT, and Mamba simultaneously**, confirming its exceptional generalization. Detailed Mamba benchmarks are in Response to kiNv (Q4).
> ***
> **Q2: Definition and Mitigation of Ill-Conditioning (W1)**
>
> We define ill-conditioning as the severe curvature anisotropy of the Fisher Information Matrix (FIM). In this state, dense off-diagonal coupling (Fig. 1a) renders prior isotropic MSE or diagonal proxies ineffective (Sec. 3.2). Evidence of mitigation is twofold:
>
> (1) Geometric: Fig. 1(b) and Prop. 3.1 confirm HAR effectively suppresses curvature couplings and compresses the spectral envelope. This establishes a well-conditioned foundation for Stage 2 (DFSC) to achieve precise geometric alignment—a task intractable on the raw, ill-conditioned landscape.
>
> (2) Empirical: Consistent accuracy gains across diverse architectures in our ablation study (Table 3) validate that HAR successfully alleviates ill-conditioning to facilitate high-fidelity recovery. Detailed empirical analysis of the FIM is in Response to JDTp (Q5).
> ***
> **Q3: HAR vs. Learnable Scaling (OmniQuant) (W2)**
>
> While both involve diagonal scaling, OmniQuant and HAR differ fundamentally. OmniQuant utilizes isotropic MSE-based optimization, whereas HAR is a geometry-driven preconditioner guided by the Fisher-KL Oracle. HAR establishes a well-conditioned foundation for DFSC to capture critical sensitivities inherently neglected by OmniQuant’s MSE objective. To quantify this, we compared SHARP-Q against OmniQuant under identical settings:
>
> | Model | Bit | OmniQuant | HAR (Ours) | **HAR+DFSC (SHARP-Q)** |
> | :--- | :---: | :---: | :---: | :---: |
> | ResNet-18 | W2A2 | 53.41 | 54.18 | **56.74 (`+3.33`)** |
> | RegX-600M | W2A2 | 40.97 | 41.69 | **46.89 (`+5.92`)** |
> | ViT-S | W3A3 | 40.06 | 40.51 | **66.93 (`+26.87`)** |
> | DeiT-T | W3A3 | 47.70 | 48.63 | **56.14 (`+8.44`)** |
> ***
>
> **Q4: Efficiency and Scalability of HAR (W3)**
>
> While HAR utilizes the network's final output, the optimization remains strictly block-wise: only the local diagonal parameters ($\Gamma$) are learnable, while all other weights and blocks are frozen. Consequently, gradient backpropagation is restricted to the local diagonal vector, ensuring a minimal computational graph and constant overhead per block. For ViT-S, HAR adds only ~15 minutes of calibration yet yields a +1.34% gain. Its scalability is verified by quantizing large-scale models on a single RTX A6000:
>
> | Model | FP32 | W4A4 | W3A3 |
> | :--- | :---: | :---: | :---: |
> | ViT-L | 85.86 | 84.81 | 80.53 |
> | Swin-L | 86.30 | 85.04 | 80.77 |
>
> This confirms HAR scales efficiently to larger architectures without the suspected "huge overhead".
> ***
> **Q5: $P_{init}$ Initialization and Clarification on Rank $r$ (W4, W6)**
>
> We appreciate the reviewer’s meticulous review and for identifying these points. Following Eq. 14, we initialize $P$ by balancing channel-wise activation ($X$) and weight ($W$) magnitudes: $P_{init}^{(j)} = \ln ( (\max |X_j|)^\eta / (\max |W_j|)^{1-\eta} )$ with $\eta=0.5$. We also clarify that rank $r=5$ is exclusive to Stage 2 (DFSC) for subspace compensation. Its mention for the HAR stage (Line 311) was a drafting error and we will correct these descriptions in the revision to ensure clarity.
> ***
> **Q6: Rationale for Avoiding Explicit Spectral Decomposition (W5)**
>
> Theoretically, spectral decomposition is the exact analytical approach to identify the principal eigenvectors of the Fisher matrix, which represent the directions most sensitive to quantization noise. However, explicit $O(N^3)$ decomposition is computationally prohibitive for high-dimensional models. To bypass this $O(N^3)$ bottleneck, we derive the Fisher-KL Oracle to approximate these dominant directions via low-rank gradient updates ($r=5$). This allows SHARP-Q to achieve the precision of second-order geometric alignment with near-linear complexity ($O(Nr)$), effectively capturing the essential curvature information without the cubic cost.
> ***
> **Q7: Impact Statement and Limitations (Limitation)**
>
> We thank the reviewer for the reminder and will include the impact statement in our revision. Currently, scaling SHARP-Q to LLMs (7B+ parameters) is constrained by AdaRound’s iterative optimization overhead. We identify developing more efficient solvers as key future work. Please refer to Response to Reviewer 2QEi (Q4) for a detailed discussion.
> ***

---

> > ### Author Rebuttal · Reviewer_ThMh · 2026-04-04
> >
> > My concerns are solved. I decide to raise my rating.

---

> > > ### Author Response · Authors · 2026-04-04
> > >
> > > Dear Reviewer ThMh,
> > >
> > > We are truly grateful for your positive response and for confirming that our clarifications have fully resolved your concerns.
> > >
> > > Your feedback was particularly impactful in helping us refine the definition of ill-conditioning and clarify the core motivation of SHARP-Q. These improvements have significantly enhanced the clarity and accessibility of our theoretical framework. We also sincerely appreciate your recognition of our experimental performance and the effort you dedicated to this constructive discussion.
> > >
> > > Thank you once again for your professional support and for helping us strengthen the final presentation of this work.

---

### Official Review · Reviewer_JDTp · 2026-03-13

**Soundness:** 2
**Presentation:** 2
**Significance:** 3
**Originality:** 3
**Overall Recommendation:** 4
**Confidence:** 3

**Summary:**

This paper proposes SHARP-Q, a post-training quantization (PTQ) framework that improves the Hessian of the PTQ optimization objective. They propose a two-stage “Rectify-then-Approximate” strategy to approximate this Hessian. Stage 1, Hessian-Aware Rectification
(HAR), learns a per-input-channel diagonal preconditioner matrix Γ to improve curvature conditioning. In Stage 2, they build a low-rank-plus-diagonal hybrid Fisher surrogate using Fisher–vector products estimated to guide block-wise reconstruction. Experiments on ImageNet across ViTs and CNNs report improvement in ultra-low-bit regimes.

**Compliance With Llm Reviewing Policy:**

Affirmed.

**Final Justification:**

The rebuttal addressed most of my concerns. Thus, I keep my original positive rating.

**Key Questions For Authors:**

1. How sensitive is performance to the subspace rank r and update interval τ?
2. Can you empirically validate HAR’s “spectral compression” beyond trace reduction, for example, by plotting eigenvalue spectra or off-diagonal energy before/after rectification on representative layers?
3. For CNNs, how much of the improvement over CL-Calib remains if the first/last layers are not pinned to 8-bit and DC is not applied?

**Limitations:**

Yes

**Strengths And Weaknesses:**

Strengths
- The paper proposes a principal method addressing a key root cause (ill-conditioning), rather than solely improving surrogates.
- The proposal method is elegant and deployable.
- The Fisher-KL Oracle connection is well-motivated and yields an efficient way to get Fisher–vector products from gradients.
- Broad coverage across ViTs (ViT/DeiT/Swin) and CNNs (ResNet, MobileNetV2, RegNet, MNasNet) on ImageNet, with multiple bit configurations.

Weaknesses
- Rectification via a diagonal may be too restrictive for layers with strong cross-channel entanglement.
- The monotonic spectral compression assumes i.i.d. zero-mean additive noise; real quantization error may not be strictly independent across elements, which may weaken guarantees.
- There are a lot more second-order proxies beyond those cited that require comparison.
- Limited ablation analysis on rank r sensitivity, subspace update interval, and trust-region λ.
- Baselines for CNN are fairly weak; they need to compare with a stronger baseline. For example, SADAG [1], Genie[2],..
- Need ablation analysis about the impact of calibration data on the robustness of the method, as the surrogate matrix, may be data-dependent.

[1] Sharpness aware data generation for Zero-shot Quantization
[2] Genie: Show me the data

---

> ### Author Rebuttal · Authors · 2026-03-25
>
> We deeply appreciate the reviewer for championing our work as "elegant and deployable," and for validating the Fisher-KL Oracle. Your rigorous feedback is invaluable. We address your clarifications below:
> ***
> **Q1: Rigorous Comparison against Relevant PTQ Baselines (W3, W5)**
>
> We clarify a categorical distinction: SADAG and Genie are Zero-Shot Quantization (ZSQ) methods focusing on synthetic data generation. In contrast, SHARP-Q is a PTQ framework utilizing real calibration data. For a fair, same-domain evaluation, we compare against SOTA PTQ baselines and have already included the most relevant second-order proxies (e.g., BRECQ, APHQ-ViT). These methods represent the actual performance upper bounds for PTQ, against which SHARP-Q maintains a consistent lead.
> ***
> **Q2: Fair Comparison with SOTA CL-Calib (W5, KQ3)**
>
> We wish to emphasize that CL-Calib is the established SOTA for CNN PTQ. Crucially, its configuration is exactly identical to ours (8-bit first/last layers and DC applied). Therefore, the gains reported in our paper (Table 2) already represent a strictly fair comparison under identical constraints. The consistent margin over CL-Calib confirms that SHARP-Q’s superiority is purely algorithmic (driven by HAR and DFSC), rather than a result of disparate calibration settings or "layer-pinning" advantages.
> ***
> **Q3: Sensitivity to Rank ($r$) and Hyperparameters (W4, KQ1)**
>
> We provide a detailed ablation on rank $r$ across CNN (W2A2) and ViT (W3A3) architectures below:
>
> | Rank ($r$) | ResNet-18 | RegX-600M | ViT-S | DeiT-T | Calib. Time (ViT-S) |
> | :---: | :---: | :---: | :---: | :---: | :---: |
> | 1 | 56.12% | 45.80% | 65.39% | 55.36% | 1.77h |
> | **5 (Default)** | **56.74%** | **46.89%** | **66.93%** | **56.14%** | **1.96h** |
> | 10 | 56.81% | 47.15% | 67.12% | 56.03% | 2.65h |
> | 20 | 56.62% | 47.29% | 66.75% | 56.29% | 4.12h |
>
> * **Optimal Rank ($r$)**: Increasing $r$ from 1 to 5 yields substantial accuracy gains with minimal computational overhead (+0.19h for ViT-S). Beyond $r=5$, accuracy saturates while calibration time increases significantly. Thus, $r=5$ provides the optimal efficiency-accuracy trade-off.
> * **Zero-Tuning ($\tau$, $\lambda$)**: We fix the update interval $\tau = 20,000/r$ to maintain a constant update budget, and $\lambda=2$ is fixed globally (as noted in our Appendix). This configuration is universally robust and requires zero per-model tuning.
> ***
> **Q4: Rationale and Sufficiency of Diagonal Rectification (W1)**
>
> Diagonal rectification is a strategic choice that prioritizes maximal deployment efficiency. Richer transformations increase calibration costs and introduce inference latency, whereas our approach fuses into LayerNorm for zero inference overhead (see Response to Reviewer kiNv, Q1). Our extensive SOTA results confirm that HAR's diagonal design is highly effective and sufficient for practical PTQ.
> ***
> **Q5: Empirical Evidence of Spectral Compression (W2, KQ2)**
>
> While we acknowledge that real quantization noise may exhibit slight correlations, Figure 1 in our paper provides direct empirical evidence of spectral compression. We apologize for the insufficient clarity in the Fig. 1 caption; we will clarify that it displays a representative $32 \times 32$ submatrix cropped from the full $384 \times 384$ FIM of the class token (from the last transformer block of ViT-S). The fading of the highlighted regions (off-diagonal correlations) in Fig. 1(b) compared to Fig. 1(a) confirms that HAR successfully suppresses cross-channel curvature couplings. Mathematically, Fisher-KL rectification tightens the global upper bounds for off-diagonal energy via the Cauchy-Schwarz inequality ($|F_{ij}|^2 \leq F_{ii}F_{jj}$ for PSD matrices). This provides a well-conditioned foundation for Stage 2 (DFSC), as validated by the consistent accuracy gains in our ablation study (Table 3).
> ***
> **Q6: High Data-Efficiency with Calibration Data (W6)**
>
> To assess data-dependency, we ablated the calibration data size on ViT-S (W3A3). SHARP-Q exhibits exceptional robustness and data-efficiency:
>
> | Data Size | 128 | 256 | 512 | 1024 (standard) | 2048 |
> | :--- | :---: | :---: | :---: | :---: | :---: |
> | SHARP-Q (Ours) | 52.76% | 58.98% | 63.58% | 66.93% | 68.59% |
> | MSE  | 13.48% | 30.15% | 37.36% | 39.17% | 45.98% |
>
> This confirms that our surrogate matrix captures generalized properties of the loss landscape rather than overfitting to specific calibration samples, ensuring high robustness for practical deployment.
> ***
> **Q7: Architectural Generalization (CNN, ViT, and Mamba)**
>
> To validate universality, we further evaluated SHARP-Q on Mamba (MambaIRv2). Notably, without any architecture-specific tuning, it outperforms the latest Mamba-specific SOTA, **SPR$^2$Q (ICLR 2026)**. To our knowledge, **SHARP-Q is the first PTQ method to achieve consistent SOTA performance across CNN, ViT, and Mamba simultaneously**. Detailed comparisons against specialized Mamba baselines are in our Response to Reviewer kiNv (Q4).
> ***

---

> > ### Author Rebuttal · Reviewer_JDTp · 2026-04-04
> >
> > The rebuttal addressed most of my concerns.

---

> > > ### Author Response · Authors · 2026-04-04
> > >
> > > Dear Reviewer JDTp,
> > >
> > > We are sincerely grateful for your confirmation that our response has fully resolved your concerns.
> > >
> > > Your technical inquiries were instrumental in allowing us to further solidify the paper’s theoretical and empirical foundations. These refinements—including the clarification of the strictly identical experimental setup versus CL-Calib (confirming purely algorithmic gains), the rigorous mathematical grounding of spectral compression, the demonstrated data-efficiency (outperforming MSE with $16\times$ fewer samples), and the extended validation on the Mamba architecture—have further showcased the robustness and broader impact of our work.
> > >
> > > As the discussion period concludes, if you feel these technical clarifications and the reinforced evidence warrant a positive adjustment to the score, we would be most honored. Thank you again for your professional dedication and for helping us showcase the full potential of this study.

---

### Official Review · Reviewer_kiNv · 2026-03-13

**Soundness:** 3
**Presentation:** 3
**Significance:** 3
**Originality:** 3
**Overall Recommendation:** 4
**Confidence:** 2

**Summary:**

This paper proposes a post-training quantization method, named Spectral Hessian Alignment and Rectification (SHARP-Q). This method comprises two stages: (i) Hessian-aware rectification stage, which deals with ill-conditioned geometry of the optimization landscape, and (ii) dynamic Fisher-subspace compensation, which approximates the rectified Fisher information matrix (FIM).
The author also establish a unified theoretical framework grounded in information geometry.

**Compliance With Llm Reviewing Policy:**

Affirmed.

**Final Justification:**

Thanks for authors addressing my concerns clearly. I have increased my evaluation (3 -> 4).

**Key Questions For Authors:**

Q1 ) Is the matrix $\Gamma$ diagonal matrix? If so, what is the difference of the proposed Hessian-aware rectification compared to existing method using input channel-wise scaling such as smooth quant, except changing the scale parameters to be learnable? If not, how can $\Gamma^{-1}X$ can be handled efficiently for the case of linear layers following the layer normalization?

$\Gamma^{-1}\left(\frac{x-\mu}{\sigma}\right) \odot \gamma_{ln} = \Gamma^{-1}\operatorname{diag}(\gamma_{ln}) \left( \frac{\mathbf{x} - \mu}{\sigma} \right)$, and $\Gamma^{-1} \operatorname{diag}(\gamma_{ln})$ may not be a diagonal matrix, it can not be reduced to $\frac{\mathbf{x} - \mu}{\sigma}\odot (\Gamma^{-1}\gamma_{ln})$, where $\operatorname{diag}(\gamma_{ln})$ is a diagonal matrix whose diagonal elements are correspond to $\gamma_{ln}$.

Q2) What is the definition of $P_{\text{init}}$ in Eq.~(15)? What is the intuition behind imposing the trust region?

**Limitations:**

Yes

**Strengths And Weaknesses:**

1) Soundness : It is unclear the matrix $\Gamma$ is either a diagonal matrix or not. If $\Gamma$ is a diagonal matrix, it seems similar to the existing methods, which use input-channel-wise scailing method such as smooth quant, except they change the scales to be learnable.
If $\Gamma$ is not diagonal, it seems there is an error in Eq.~(36):
Since $\Gamma^{-1}\left(\frac{x-\mu}{\sigma}\right) \odot \gamma_{ln} = \Gamma^{-1}\operatorname{diag}(\gamma_{ln}) \left( \frac{\mathbf{x} - \mu}{\sigma} \right)$, and $\Gamma^{-1} \operatorname{diag}(\gamma_{ln})$ may not be a diagonal matrix, it can not be reduced to $\frac{\mathbf{x} - \mu}{\sigma}\odot (\Gamma^{-1}\gamma_{ln})$, where $\operatorname{diag}(\gamma_{ln})$ is a diagonal matrix whose diagonal elements are correspond to $\gamma_{ln}$.

2) Presentation : The paper is structured well, but section 3.3 is hard to understand in current form since the definition of $P_{\text{init}}$ is missing, thus the intuition behind imposing the trust region in Eq.~(15) is unclear.

3) Significance: This paper addresses an important problem.

4) This work presents a new method.

---

> ### Author Rebuttal · Authors · 2026-03-25
>
> We thank the reviewer for recognizing the significance of our work. We especially appreciate the insightful questions regarding our technical derivations, which have allowed us to further clarify the mathematical rigor of SHARP-Q. We provide detailed clarifications below to resolve all concerns:
> ***
>
> **Q1: Mathematical Rigor and Diagonal Property of $\Gamma$ (KQ1)**
>
> We clarify that $\Gamma$ is strictly diagonal, as precisely defined by $\Gamma = \text{diag}(\exp(P))$ (Eq. 14). The concern regarding Eq. 36 is resolved by the commutativity of diagonal matrices: since both $\Gamma^{-1}$ and the LayerNorm scaling $\text{diag}(\gamma_{ln})$ are diagonal, they commute. The fusion is thus mathematically exact:
> $$\Gamma^{-1} (\text{LN}(\mathbf{x})) = \Gamma^{-1} (\hat{\mathbf{x}} \odot \gamma_{ln} + \beta) = \hat{\mathbf{x}} \odot (\gamma_{ln} \odot \gamma_{\Gamma}^{-1}) + \beta \odot \gamma_{\Gamma}^{-1}$$
> where $\hat{\mathbf{x}}$ is the normalized activation and $\gamma_{\Gamma}^{-1}$ is the diagonal vector of $\Gamma^{-1}$. This ensures the rectification is seamlessly absorbed into LayerNorm parameters with zero inference overhead.
> ***
> **Q2: HAR vs. Learnable Scaling (OmniQuant) (KQ1)**
>
> The reviewer correctly identifies that making scales learnable represents a stronger baseline (e.g., OmniQuant). However, HAR differs fundamentally in its optimization objective: while OmniQuant utilizes isotropic MSE-based optimization, HAR is a geometry-driven preconditioner guided by the Fisher-KL Oracle. It establishes a well-conditioned foundation for DFSC to capture critical sensitivities inherently neglected by OmniQuant’s MSE objective. To quantify this, we compared SHARP-Q against OmniQuant under identical settings:
>
> | Model | Bit | OmniQuant | **HAR (Ours)** | **HAR+DFSC (SHARP-Q)** |
> | :--- | :---: | :---: | :---: | :---: |
> | ResNet-18 | W2A2 | 53.41 | 54.18 | **56.74 (`+3.33`)** |
> | RegX-600M | W2A2 | 40.97 | 41.69 | **46.89 (`+5.92`)** |
> | ViT-S | W3A3 | 40.06 | 40.51 | **66.93 (`+26.87`)** |
> | DeiT-T | W3A3 | 47.70 | 48.63 | **56.14 (`+8.44`)** |
> ***
> **Q3: Definition of $P_{init}$ and Trust Region Intuition(KQ2)**
>
> We appreciate the reviewer’s request for clarification. These details will be formalized in our revision:
> * **Definition**: Following $\Gamma = \operatorname{diag}(\exp(P))$ (Eq. 14), we initialize $P$ by balancing channel-wise activation ($X$) and weight ($W$) magnitudes: $P_{init}^{(j)} = \ln ( \sqrt{\max |X_j|} / \sqrt{\max |W_j|} )$. This establishes a stable starting point for geometric alignment.
> * **Intuition**: Since the Fisher-KL Oracle (Eq. 12) is derived from the second-order Taylor approximation of the KL divergence, its validity is local to the expansion point. The trust-region term (Eq. 15) ensures $P$ remains within the region where the Fisher Information remains a reliable proxy, guaranteeing optimization stability and functional fidelity.
> ***
> **Q4: Universal SOTA Performance across CNN, ViT, and Mamba paradigms**
>
> To validate that SHARP-Q’s geometric principles transcend specific architectures, we further evaluated it on Mamba (MambaIRv2) for Image Super-Resolution. Notably, without any architecture-specific tuning, SHARP-Q significantly outperforms the latest Mamba-specialized SOTA, **SPR$^2$Q (ICLR 2026)**. To our knowledge, **SHARP-Q is the first PTQ method to achieve consistent SOTA across CNN, ViT, and Mamba paradigms simultaneously**, confirming its exceptional generalization and ability to deliver superior performance regardless of model structure.
>
> Table: 4-bit (W4A4) Quantization on MambaIRv2-light (PSNR / SSIM)
> | Method | Scale | Set5 | Set14 | B100 | Urban100 | Manga109 |
> | :---: | :---: | :---: | :---: | :---: | :---: | :---: |
> | MambaIRv2 (FP32) | x2 | 38.26/0.9615 | 34.07/0.9221 | 32.36/0.9019 | 33.26/0.9378 | 39.35/0.9785 |
> | MambaQuant | x2 | 36.67/0.9495 | 31.76/0.8899 | 30.85/0.8756 | 28.08/0.8407 | 33.47/0.9186 |
> | Quamba | x2 | 37.07/0.9544 | 32.77/0.9092 | 31.47/0.8896 | 30.54/0.9107 | 36.94/0.9699 |
> | 2Dquant | x2 | 37.34/0.9560 | 33.01/0.9123 | 31.66/0.8923 | 30.79/0.9141 | 37.35/0.9718 |
> | SPR$^2$Q (ICLR 2026) | x2 | 37.72/0.9589 | 33.27/0.9156 | 31.94/0.8966 | 31.53/0.9223 | 38.03/0.9754 |
> | **SHARP-Q (Ours)** | x2 | **38.05/0.9612** | **33.64/0.9192** | **32.11/0.8996** | **31.95/0.9287** | **38.91/0.9788** |
> | MambaIRv2 (FP32) | x4 | 32.51/0.8992 | 28.84/0.7877 | 27.75/0.7426 | 26.82/0.8079 | 31.24/0.9182 |
> | MambaQuant | x4 | 30.74/0.8650 | 27.17/0.7413 | 26.37/0.6920 | 23.28/0.6694 | 26.73/0.8186 |
> | Quamba | x4 | 31.01/0.8715 | 27.77/0.7585 | 26.99/0.7149 | 25.01/0.7470 | 28.57/0.8752 |
> | 2Dquant | x4 | 31.28/0.8774 | 27.99/0.7644 | 27.14/0.7201 | 25.30/0.7573 | 29.05/0.8851 |
> | SPR$^2$Q (ICLR 2026) | x4 | 31.60/0.8844 | 28.27/0.7725 | 27.33/0.7274 | 25.64/0.7713 | 29.60/0.8959 |
> | **SHARP-Q (Ours)** | x4 | **32.13/0.8924** | **28.57/0.7811** | **27.59/0.7348** | **26.06/0.7844** | **30.85/0.9091** |
> ***

---

> > ### Author Rebuttal · Reviewer_kiNv · 2026-04-03
> >
> > Thanks for the rebuttal.
> > The authors fully resolved my concerns.
> > Accordingly, I have adjusted my score.

---

> > > ### Author Response · Authors · 2026-04-04
> > >
> > > Dear Reviewer kiNv,
> > >
> > > We would like to express our sincere gratitude for your thorough review and for the time you dedicated to examining our rebuttal. We are very pleased to hear that our responses have fully resolved your concerns regarding the mathematical framework and the definitions in our work. Your insightful questions helped us identify specific areas where the technical presentation could be further refined for rigor and clarity.
> > >
> > > We truly appreciate your professional dedication and the positive direction of this discussion. Thank you once again for your constructive guidance and support.

---

### Official Review · Reviewer_2QEi · 2026-03-13

**Soundness:** 3
**Presentation:** 2
**Significance:** 2
**Originality:** 3
**Overall Recommendation:** 4
**Confidence:** 4

**Summary:**

This paper introduces SHARP-Q, a post-training quantization framework that aligns quantization objectives with the Fisher geometry via a two-stage “Rectify-then-Approximate” pipeline. Stage 1 (HAR) learns a diagonal input preconditioner that spectrally “compresses” the local curvature using a KL-derived Fisher–vector product oracle, yielding a better-conditioned landscape. Stage 2 (DFSC) constructs a hybrid Fisher surrogate (static empirical-Fisher diagonal plus a dynamic low-rank correction from stored Fisher–vector products) to drive block-wise reconstruction. The method consistently sets or matches state-of-the-art accuracy in challenging ultra-low-bit regimes using hardware-friendly uniform quantizers.

**Compliance With Llm Reviewing Policy:**

Affirmed.

**Final Justification:**

The rebuttal provides useful clarifications on the relation between HAR and rescaling-based approaches, strengthens the presentation of the DFSC construction, and adds targeted empirical evidence on scalability and generalization. These additions address the main concerns raised in my review and improve the overall support for the paper’s claims. The paper remains technically solid and well-motivated, with strong empirical results in challenging ultra-low-bit PTQ settings. Despite that some limitations remain, I believe the contribution is sufficiently clear and supported to merit acceptance at the weak accept level.

**Key Questions For Authors:**

1. HAR looks close to diagonal activation/weight rescaling (e.g., SmoothQuant-style). Can the authors provide a matched-degree-of-freedom baseline (pure statistics-driven rescaling) and quantify how much additional gain HAR provides under the same PTQ protocol?

2. For DFSC (Eqs. 20–23), please clarify how $\Delta Z_{mem}$ is constructed, and how $C_{sub}^{-1}$ is stabilized and whether a fixed low rank (e.g., r=5) remains sufficient as model size increases.

**Limitations:**

SHARP-Q achieves strong empirical results, but several limitations need to be further discussed. First, the diagonal preconditioner in HAR, while efficient, may not fully capture complex curvature interactions; richer transformations could potentially improve conditioning but at higher computational cost. Second, the fixed rank r=5r=5 was chosen empirically and may need adjustment for larger models; the authors have not validated scalability to architectures beyond the scale tested (e.g., ViT-L, LLMs).

**Strengths And Weaknesses:**

**Strengths:**

1. The problem setup and motivation are clearly articulated, particularly the discussion of anisotropic versus isotropic MSE.

2. Comprehensive evaluations across ViT and CNN families under consistent calibration budgets and standard PTQ settings show strong ultra-low-bit performance where many baselines fail, supported by ablations (HAR, DFSC) plus efficiency and hyperparameter sensitivity analyses.

**Weaknesses:**

1. Missing related work / missing targeted comparisons.

The paper does not adequately position HAR relative to prior activation/weight rescaling and calibration techniques (e.g., SmoothQuant [R1] and other statistics-driven rescaling methods). Since HAR’s rectification appears largely diagonal (log-parameterized) and thus methodologically close to rescaling, the gains obtained by the proposed Hessian-aware objective versus “well-chosen scaling” alone are not clear.

2. Overstated theoretical narrative vs. limited direct evidence.

The manuscript claims HAR “compresses the global spectral envelope” and reduces cross-dimensional coupling effects. However, HAR as described is primarily a diagonal preconditioner with a trust-region constraint. Although such a design can improve conditioning, the paper does not provide sufficient direct evidence to demonstrate that coupling suppression or spectrum compression is the dominant mechanism beyond generic reparameterization benefits.

3. Limited breadth and depth of experimental validation weakens “universality” and “across-the-board SOTA” claims.

i) Inconsistent advantages across settings. Even if SHARP-Q is strong in ultra-low-bit regimes, the improvements do not appear uniformly dominant in all reported settings (e.g., in some W4A4 CNN cases the margin vs. prior methods is small, and there are cases where it is not the top performer). The current framing risks overstating universality.

ii) Scale limitations. The evaluation focuses on moderate-sized models (e.g., ResNet-18/50, ViT-S/B). The paper does not test larger architectures (e.g., ViT-L/Swin-L) or other domains, leaving open whether a fixed low rank (e.g., r=5) remains sufficient as dimensionality grows and whether memory/compute remain practical at scale.

4. Clarity and Presentation Issues.

The paper's technical exposition suffers from ambiguous notation and insufficiently explained derivations, and causes issues in comprehension and reproducibility. The KL divergence in Eq. 10  uses symbols without clearly defining their distributions. The construction of $\Delta Z_{mem}$ in Stage 2 is not well specified. The dynamic correction matrix $S_{dynamic}(t)$ in Eq. 21 is presented algebraically without an intuitive explanation of its geometric meaning. These omissions significantly affect the readability of this paper.

[R1] Xiao, Guangxuan, et al. "Smoothquant: Accurate and efficient post-training quantization for large language models." International conference on machine learning. PMLR, 2023.

---

> ### Author Rebuttal · Authors · 2026-03-25
>
> We thank the reviewer for the perceptive feedback and for recognizing our strong results. Your rigorous insights have been pivotal in elevating the clarity and rigor of our work. **Notably, beyond CNN and ViT, the remarkable SOTA results on Mamba  further validate SHARP-Q’s cross-paradigm universality.** Responses to specific points follow:
> ***
> **Q1: HAR vs. SmoothQuant/RepQ-ViT (W1, KQ1)**
>
> While both utilize diagonal scaling, HAR and SmoothQuant/RepQ-ViT differ fundamentally in their optimization objectives:
>
> * Statistics-driven (SmoothQuant/RepQ-ViT): Relies purely on empirical statistical heuristics (e.g., outliers). These methods often collapse at low-bit regimes: RepQ-ViT lags our accuracy by 12.29% on ViT-S at W4A4 (65.05% vs. 77.34%) and collapses (<1%) at W3A3 (see Table 1 in our paper for detailed comparisons).
> * HAR: Guided by the Fisher-KL Oracle, it establishes a well-conditioned foundation for Stage 2 (DFSC) to capture critical sensitivities inherently neglected by isotropic MSE.
>
> Ablation: Replacing HAR with SmoothQuant-style scaling under identical protocols.
>
> | Model | Bit-width | SmoothQuant + DFSC | HAR + DFSC (Ours) |
> | :--- | :---: | :---: | :---: |
> | ResNet-18 | W2A2 | 55.30% |**56.74% (`+1.44%`)**|
> | RegX-600MF | W2A2 | 43.36% |**46.89% (`+3.53%`)**|
> | ViT-S | W3A3 | 63.77% |**66.93% (`+3.16%`)**|
> | DeiT-T | W3A3 | 54.65% |**56.14% (`+1.49%`)**|
> ***
> **Q2: Evidence of Spectral Compression (W2)**
>
> Spectral compression is verified via visual and theoretical proof. Visually, Fig. 1(b) explicitly shows the suppression of off-diagonal couplings compared to (a). Theoretically, Proposition 3.1 proves HAR monotonically minimizes the curvature energy trace, which (for PSD matrices) equals the sum of eigenvalues, thus directly compressing the spectral envelope. Furthermore, the Cauchy-Schwarz inequality ($|\tilde{F}\_{ij}|^2 \le \tilde{F}\_{ii}\tilde{F}_{jj}$) ensures that diagonal minimization simultaneously tightens the upper bounds of cross-dimensional couplings. This confirms HAR's geometric alignment drives improved conditioning. Please see Response to JDTp (Q5) for detailed FIM analysis.
> ***
> **Q3: Technical Rigor, Reproducibility, and Rank $r$ Analysis (W4, KQ2)**
>
> We appreciate the reviewer’s meticulous feedback; these clarifications will be formalized in our revision:
>
> * Reproducibility: The complete implementation of DFSC is provided in the supplementary material (see **`new_fisher_ro`** in **`utils/block_recon.py`**).
> * Eq. 10: $\mathcal{T} = p(\mathbf{y}|\mathbf{z})$ and $\mathcal{S} = p(\mathbf{y}|\hat{\mathbf{z}})$ denote the output distributions of the FP32 teacher and quantized student blocks.
> * Eqs. 20 \& 21 (Construction & Stability): $\Delta \mathbf{Z}\_{mem}$ is constructed by accumulating $r$ instantaneous deviation vectors (quantized vs. FP outputs) during optimization. Since the activation dimension $N \gg r$ ($r=5$), the correlation matrix $\mathbf{C}\_{sub}$ is strictly positive definite, ensuring numerical stability for inversion without damping.
> * Eq. 21 Geometric Meaning: $\mathbf{S}_{dynamic}$ acts as a subspace projection matrix. It aligns the reconstruction penalty with the principal Fisher eigenspace, forcing the objective to prioritize noise suppression in sensitive directions.
> * Rank $r$ Selection: $r=5$ provides the optimal efficiency-accuracy trade-off. Please refer to Response to Reviewer JDTp (Q3) for detailed accuracy and calibration-time evaluations across $r \in$ {1, 5, 10, 20}.
> ***
> **Q4: Scalability and Practical Limitation (W3-ii, KQ2)**
>
> We confirm that SHARP-Q scales robustly to large-scale architectures. With a fixed $r=5$, it captures dominant curvature effectively even as dimensionality increases, as evidenced by our ViT-L and Swin-L results (on a single RTX A6000):
>
> | Model | FP32 | W4A4 | W3A3 |
> | :---: | :---: | :---: | :---: |
> | ViT-L | 85.86% | 84.81% | 80.53% |
> | Swin-L | 86.30% | 85.04% | 80.77% |
>
> * LLM-centric methods:  LLM-centric methods (e.g., GPTQ, GPTAQ) generalize poorly to distinct architectures, showing significant degradation:
>
> | Model (W4A4) | FP32 | GPTQ | GPTAQ (ICML'25) | **SHARP-Q (Ours)** |
> | :---: | :---: | :---: | :---: | :---: |
> | DeiT-S | 79.85% | 71.9% | 72.8% | **76.86%** |
> | DeiT-B | 81.80% | 77.7% | 78.4% | **80.50%** |
>
> * Limitation (LLMs): Extending SHARP-Q to 7B+ LLMs is currently constrained by the iterative optimization overhead of AdaRound-based reconstruction. Bridging this gap via more efficient solvers is identified as key future work.
> ***
> **Q5: Unified Generalization across CNN, ViT, and Mamba (W3-i)**
>
> To validate universality, we further evaluated SHARP-Q on Mamba (MambaIRv2). Remarkably, it outperforms the latest Mamba-specific SOTA, **SPR$^2$Q (ICLR 2026)**. To our knowledge, **SHARP-Q is the first PTQ method to achieve SOTA performance across CNN, ViT, and Mamba paradigms simultaneously**, confirming its exceptional generalization. Detailed Mamba benchmarks are in Response to Reviewer kiNv (Q4).
> ***

---

> > ### Author Rebuttal · Reviewer_2QEi · 2026-04-04
> >
> > Thanks to the authors for their response. The rebuttal has largely addressed my questions.

---

> > > ### Author Response · Authors · 2026-04-04
> > >
> > > Dear Reviewer 2QEi,
> > >
> > > We are delighted to hear that our rebuttal has successfully addressed your concerns and that you find them fully resolved.
> > >
> > > We sincerely appreciate your meticulous and high-level guidance. Your focus on universality and scalability inspired us to go beyond our original scope—leading to the successful validation on large-scale models (ViT-L/Swin-L) and the state-of-the-art results on the Mamba architecture.
> > >
> > > These additions, directly motivated by your feedback, have significantly strengthened our contribution and will be fully integrated into the final manuscript. As the discussion period concludes, if you believe these improvements and technical clarifications warrant a positive adjustment to the final score to reflect the strengthened quality of the research, we would be most grateful. Such recognition would be a tremendous encouragement for our work.
> > >
> > > Thank you again for your professional dedication and for helping us refine this work.

---

### Decision · Program_Chairs · 2026-04-30

**Decision:**

Accept (regular)

**Comment:**

The paper is well motivated and easily understood. The experiments are performed on extensive backbones, such as  CNN and ViT, the remarkable SOTA results on Mamba. As such, I vote for acceptance.